# A cluster of broadly neutralizing IgG against BK polyomavirus in a repertoire dominated by IgM

Ngoc-Khanh Nguyen[1], Marie-Claire Devilder[2], Laetitia Gautreau-Rolland[2,3], Cynthia Fourgeux[1], Debajyoti Sinha[1], Jeremie Poschmann[1], Maryvonne Hourmant[5], Céline Bressollette-Bodin[1,4,6], Xavier Saulquin[2,3,*], Dorian McIlroy[1,3,*]

The BK polyomavirus (BKPyV) is an opportunistic pathogen, which is only pathogenic in immunosuppressed individuals, such as kidney transplant recipients, in whom BKPyV can cause significant morbidity. To identify broadly neutralizing antibodies against this virus, we used fluorescence-labeled BKPyV virus-like particles to sort BKPyV-specific B cells from the PBMC of KTx recipients, then single-cell RNAseq to obtain paired heavy- and light-chain antibody sequences from 2,106 sorted B cells. The BKPyV-specific repertoire was highly diverse in terms of both V-gene usage and clonotype diversity and included most of the IgM B cells, including many with extensive somatic hypermutation. In two patients where sufficient data were available, IgM B cells in the BKPyV-specific dataset had significant differences in V-gene usage compared with IgG B cells from the same patient. CDR3 sequence–based clustering allowed us to identify and characterize three broadly neutralizing "41F17-like" clonotypes that were predominantly IgG, suggesting that some specific BKPyV capsid epitopes are preferentially targeted by IgG.

## Introduction

The BK polyomavirus (BKPyV) is a typical opportunistic pathogen. After asymptomatic primary infection during childhood, it establishes a latent infection in the kidney, which appears to persist throughout life. ~7% of healthy adults excrete BKPyV in the urine [1], and this proportion increases during acquired [2] or iatrogenic [3] immunosuppression. Its pathogenic potential is manifested in patients treated with allogeneic hematopoietic stem-cell transplantation, in whom BKPyV replication can cause hemorrhagic cystitis (BKPyV-HC), and kidney transplant (KTx) recipients, in whom uncontrolled BKPyV replication can result in polyomavirus nephropathy (PyVAN) and graft loss or dysfunction. PyVAN can only be

diagnosed definitively by histology, but it is correlated with DNAemia greater than $10^4$ genome copies/ml [4], and high-level DNAemia is generally classified as presumptive PyVAN [5]. There is currently no approved antiviral therapy with clinical efficacy against BKPyV, so presumptive or biopsy-confirmed PyVAN is managed by modulation of immunosuppressive therapy, which allows host immune responses to clear the virus [6]. The virological response rate to this intervention appears to vary between centers, with recent publications reporting clearance of DNAemia in response to modulation of immunosuppression in proportions varying from 30% [7], [8] up to more than 75% [9] of PyVAN patients. In the single-center study with the longest follow-up and the largest cohort, at least 25% of PyVAN patients had DNAemia that persisted for more than 1 yr, despite modulation of immunosuppressive therapy [10]. Similarly, the Banff working group on PyVAN, analyzing data from nine transplant centers in Europe and North America, found that PyVAN persisted for more than 24 mo in 39 of 149 (26%) patients [11]. Importantly, recent analysis indicates that persistent PyVAN is associated with an increased risk of graft failure and that graft loss occurs almost exclusively in patients with persistent PyVAN [12].

Because polyomavirus replication does not involve viral enzymes, there is no obvious target for small-molecule antivirals, and this has led to the exploration of immunotherapies as potential treatment options for patients with PyVAN. Several previous studies have shown that the antiviral CTL response plays a key role in BKPyV clearance [13], [14], [15], leading to the development of cellular immunotherapy for active BKPyV replication in the context of hematopoietic stem-cell transplantation [16], [17] and solid organ transplant [18]. With respect to the BKPyV-specific humoral response, PyVAN is reported to occur more frequently in KTx recipients with low ELISA [19] and neutralizing antibody titers [20]. Furthermore, some clinical studies and case reports have indicated that infusion of intravenous immunoglobulin-containing BKPyV-specific neutralizing antibodies [21] can prevent active BKPyV replication in KTx recipients [22] and successfully treat PyVAN [23], [24], [25].

These encouraging observations have stimulated the search for monoclonal antibodies against BKPyV, specifically those with broad

---

[1]Nantes Université, CHU Nantes, INSERM, Center for Research in Transplantation and Translational Immunology, UMR 1064, ITUN, Nantes, France  [2]Nantes Université, Inserm UMR 1307, CNRS UMR 6075, Université d'Angers, CRCI2NA, Nantes, France  [3]UFR Sciences et Techniques, Nantes Université, Nantes, France  [4]CHU Nantes, Nantes Université, Service de Virologie, Nantes, France  [5]CHU Nantes, Nantes Université, Service de Néphrologie-Immunologie clinique, Nantes, France  [6]UFR Médecine, Nantes Université, Nantes, France

Correspondence: xavier.saulquin@univ-nantes.fr; dorian.mcilroy@univ-nantes.fr
*Xavier Saulquin and Dorian McIlroy contributed equally to this work

---

 

specificity, capable of neutralizing all four BKPyV genotypes ([26]). Indeed, two such antibodies are currently in clinical trials (NCT04294472 and NCT05358106). However, the robust humoral response after BKPyV reactivation does not always coincide with control of virus replication ([15], [19], [27]), and persistent BKPyV DNAemia in the face of a strong humoral response appears to be related to the emergence of neutralization escape mutations in the virus capsid ([28], [29]). This raises the question of whether the clinical efficacy of single monoclonal antibodies might be limited by the emergence of viral escape mutations.

In the present work, we aimed to isolate broadly neutralizing antibodies from KTx recipients who had successfully controlled BKPyV replication, and tested selected antibodies, and combinations of pairs of antibodies, against a panel of neutralization escape mutants, in order to define a broadly neutralizing antibody cocktail that retained its antiviral activity against neutralization escape mutants in both a genotype I (gI) and a genotype IV (gIV) background. The experimental approach that we adopted, combining sorting of specific B cells with fluorescence-labeled BKPyV virus-like particles (VLPs) and single-cell RNA sequencing (scRNAseq), followed by expression and characterization of selected antibodies, also generated data on the B-cell receptor (BCR) repertoire and gene expression profile in KTx recipients' circulating BKPyV-specific B cells. A surprising feature of these data was the high frequency of the IgM isotype in BCR sequences from sorted B cells, which confirms previous observations in healthy donors ([26]) and complements recent reports on the role of memory IgM responses in viral ([30]) and other infections ([31]).

# Results

## Sorting of BKPyV-specific B cells from KTx recipients

Fluorescence-labeled BKPyV and MPyV VLPs were prepared by transient expression of VP1 in HEK293TT cells, followed by labeling of cell lysates and purification of VLPs by ultracentrifugation. VLP morphology and purity were confirmed by negative stain electron microscopy (Fig 1A) and SDS–PAGE (Fig 1B), and no serological cross-reactivity was found between BKPyV VLPs and MPyV VLPs (Fig 1C).

Preliminary experiments staining PBMC with labeled VLPs showed that BKPyV-specific B cells represented at least 0.2% of CD19$^+$ B cells in KTx recipient 3.12, who was seropositive and experienced high BKPyV viruria and viremia. In contrast, in healthy donors and BKPyV-seronegative KTx with no evidence of viral reactivation, BKPyV-specific B cells were extremely rare, comprising no more than 0.08% of the CD19$^+$ population (Fig S1). Furthermore, BKPyV-specific antibodies were successfully cloned and expressed after sorting single B cells double-positive for BKPyV VLPs and negative for MPyV VLPs (Fig S2), confirming that fluorescent BKPyV VLPs could be used as probes to sort circulating BKPyV-specific B cells.

In order to obtain a sufficient number of BKPyV-specific B cells for scRNAseq, we developed a strategy to process multiple frozen PBMC samples on the same day (Fig 1D). Six KTx patients who had significant increases in neutralizing antibody titers after BKPyV reactivation (Fig S3) were selected for this experiment. A total of 270

million frozen PBMCs from 17 samples (at least two independent PBMC samples per patient) were thawed, and then, B cells were enriched from each sample by magnetic sorting. Enriched B cells were stained in parallel with antibodies against CD3-BV510–, CD19-BV421–, AF555-, and AF647-labeled BKPyV VLP gIa, and MPyV VLP-AF488, in addition to two oligonucleotide hashtagged antibodies for each sample (Fig S4). After separately processing and staining each PBMC sample, labeled B cells were pooled and FACS-sorted. To study the total B-cell repertoire, roughly 10$^5$ CD19$^+$ B cells were isolated first (TotB population), and then, the BKPyV-specific B cells were sorted from the remainder of the sample, yielding ~10$^4$ sorted cells (SpecB population) (Fig 1E). The purity of sorted populations was validated by re-analyzing a small proportion of the sorted cells (Fig S5). This showed that ~90% of sorted BKPyV-specific B cells stained double-positive for AF555- and AF647-labeled BKPyV VLPs, and negative for AF488-labeled MuPyV VLPs. Subsequent to FACS sorting, ~6 × 10$^4$ TotB and 1 x 10$^4$ SpecB cells were processed on the Chromium 10× platform to generate single-cell cDNA libraries for (1) heavy- and light-chain BCR sequences, (2) 5′ gene expression profiles, and (3) antibody hashtags identifying the sample. Libraries were pooled for the TotB and the SpecB sample, then sent for Illumina sequencing. The number of PBMC, enriched B cells, and antibody sequences recovered for each patient sample is detailed in Table S1.

## BCR repertoire of BKPyV-specific B cells

Paired heavy- and light-chain BCR sequences (in AIRR format in Table S2) were obtained for 2,106 sorted BKPyV-specific B cells, and 4,591 total B cells. The distribution of BKPyV-specific antibody sequences was very uneven between patients. More than 1,500 antibody sequences were derived from patient 3.1, whereas only 13 antibody sequences from patient 3.12 and only 12 from patient 3.3 were obtained. However, at least 50 BKPyV-specific BCR sequences were obtained from each of the other three patients (Fig 2A), in addition to several hundred paired heavy- and light-chain BCR sequences from circulating total CD19$^+$ B cells from the same patients (Fig 2A). In all four patients with at least 50 BKPyV-specific antibody sequences, most BKPyV-specific B cells carried lambda light chains, whereas the total B-cell repertoire from the same patients was dominated by antibodies with k light chains (Fig 2B). The proportion of IgD and IgA antibodies was lower, and the proportion of IgG antibodies was higher in BKPyV-specific B cells compared with the total B-cell population. Nevertheless, in all four patients, IgM was the majority isotype in BKPyV-specific and total B cells (Fig 2C). In terms of V-gene usage, all four patients showed a greater than twofold enrichment of IGHV4-39 in BKPyV-specific B cells compared with total B cells from the same patient, and in patients 3.1, 3.4, and 2.6, IGHV4-39 was the dominant heavy-chain V-gene, present in at least 20% of BKPyV-specific B cells (Fig S6). In terms of light-chain V-gene usage, no single V-gene stood out, although IGLV2-11 showed a greater than twofold enrichment in three of the four patients.

Clonotype diversity, as measured by the Shannon diversity and the D50 index (Fig 2D and E), was significantly lower in BKPyV-specific B cells compared with the total B-cell population, whereas

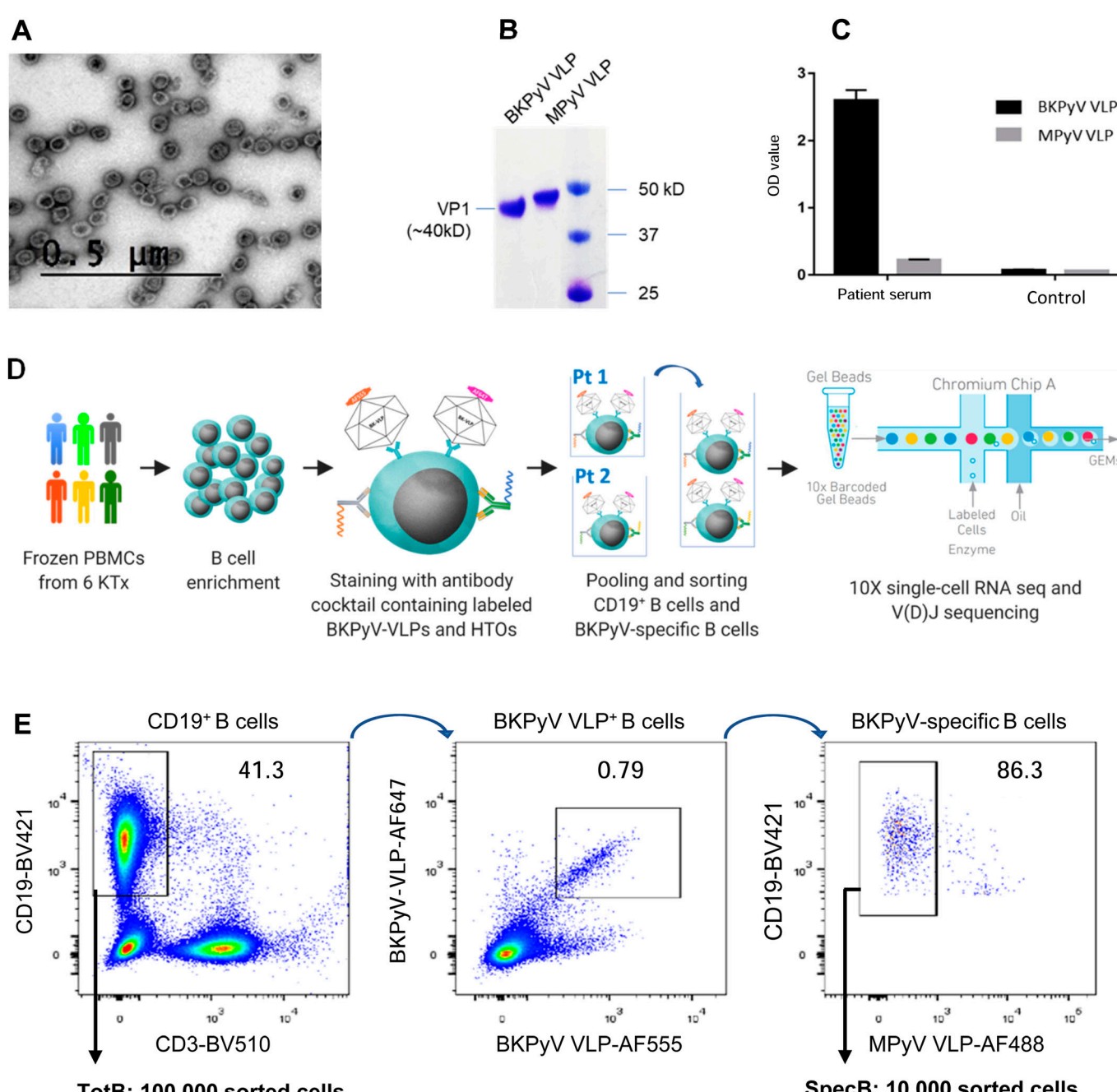

**Figure 1. Fluorescence-labeled VLP characterization and isolation of BKPyV-specific B cells.**
**(A)** Negative stain electron microscopy of fluorescent VLPs. Bar = 0.5 μm. **(B)** Fluorescence-labeled BKPyV VLP and MPyV VLP on SDS–PAGE protein gel. **(C)** IgG ELISA binding of serum from KTx recipient 3.1 and a seronegative KTx recipient to BKPyV and MPyV VLPs. **(D)** Strategy of the scRNAseq experiment. HTO, hashtag oligonucleotide. **(E)** FACS profile of pooled BKPyV-specific B cells indicating the SpecB and TotB populations that were sorted.

the unevenness, or inequality, of the repertoire measured by the Gini coefficient (Fig 2F) was significantly higher. These characteristics are expected for antigen-specific B-cell repertoires: diversity (as measured by the D50 and the Shannon indices) is lower, because the number of antigen-specific clonotypes is smaller than the total B-cell repertoire, whereas the expansion of dominant clones generates inequality in antigen-specific repertoires, leading to a higher Gini coefficient.

Having access to the BCR repertoire in total B cells and the sorted BKPyV-specific B cells allowed us to test directly the possibility that antibody sequences in the SpecB dataset may have been contaminated by sequences from the total B-cell population. In patient 3.1, the TotB dataset contained one dominant clone (clone ID 4968), represented by 46 cells, and 21 expanded clones with at least three cells in the dataset per clonotype. In the SpecB dataset from the same patient, clone 4,968 was not found, and of the 21 expanded

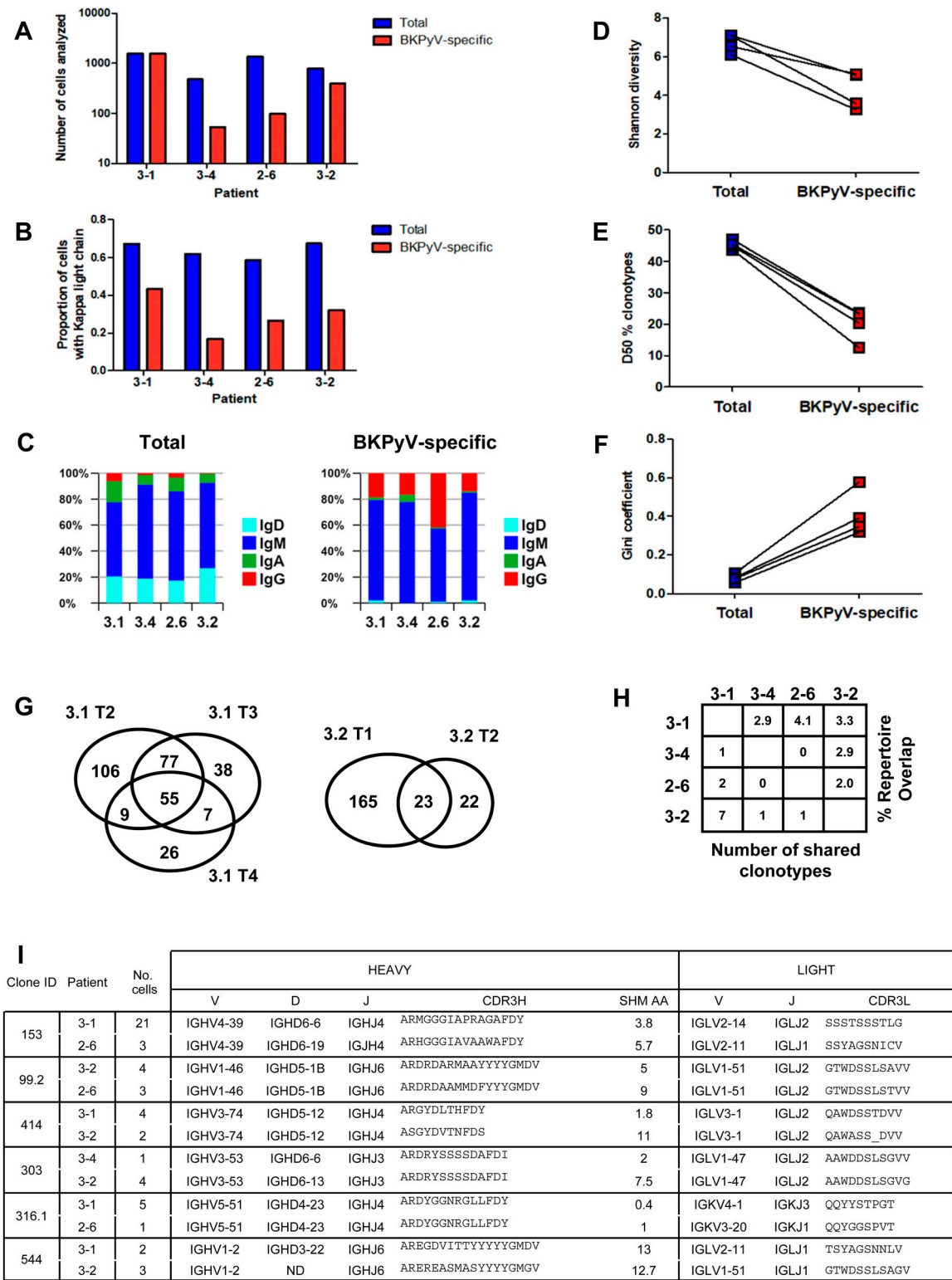

**Figure 2. Features of BKPyV-specific BCR repertoire.**
**(A)** Number of cells with paired heavy- and light-chain antibody sequences obtained per patient from sorted BKPyV-specific (red) and total B cells (blue). **(B)** Proportion of cells expressing kappa light-chain antibodies in BKPyV-specific and total B cells. **(C)** Proportion of cells expressing different antibody isotypes in total (left panel) and BKPyV-specific (right panel) B cells. **(D, E, F)** Shannon diversity, D50 index, and Gini coefficient of clonotype distributions observed in BKPyV-specific and total B cells. Paired data represent cells from the same patient. **(G)** Venn diagram of B-cell repertoire overlap in BKPyV-specific cells observed in independent PBMC samples from patients 3.1 and 3.2. Numbers represent the number of clonotypes observed in each sector of the graph, independent of the number of cells per clonotype. **(H)** B-cell repertoire overlap in SpecB cells between patients. The percentage overlap is calculated as the number of shared clonotypes divided by the number of clonotypes in the smaller dataset. **(I)** Sequence characteristics of selected "public" clonotypes.

clones, only three (clone IDs 152, 81, and 304.1) were represented in the SpecB dataset, where they were among the top 5 expanded clones. The specificity of these three clones was validated experimentally (Fig 6). The expanded clonotypes in the TotB dataset that were not detected in the SpecB dataset (and therefore non-specific for BKPyV) made up 6.2% of the TotB BCR repertoire (105 cells of 1,691), and this allowed us to estimate the potential contamination of the 1,542 SpecB BCR sequences with non-specific sequences. For example, if the SpecB repertoire contained 10% sequences from total B cells, then we would expect that 0.6% of the SpecB repertoire, or 10 cells, would carry BCR sequences from the non-specific expanded clonotypes in the TotB repertoire. Because this was significantly different ($P < 0.001$, Fisher's exact test) from the observed number (i.e., zero cells), we can therefore conclude that the contamination of the SpecB dataset with non-specific antibody sequences was less than 10%, and possibly less than 5% ($P < 0.05$), for patient 3.1. As the cells for the other patients were sorted at the same time, this estimate is likely to be valid for the other patients, and indeed, the same analysis carried out on clonotypes found in SpecB and TotB datasets for patient 3.2 gave similar results (Table S3).

Sufficient data were available for patients 3.1 and 3.2 to analyze the BKPyV-specific repertoire overlap in independent PBMC samples from the same individual (Fig 2G). The BKPyV-specific repertoire overlap found in two samples from patient 3.2, calculated as the number of shared clonotypes divided by the number of clonotypes in the smaller dataset, was 51%, whereas for the three PBMC samples that provided sufficient data from patient 3.1, three independent two-by-two measures of repertoire overlap were obtained (3.1 T2 versus 3.1 T3: 132/177 = 74.6%; 3.1 T2 versus 3.1 T4: 64/97 = 66.0%; 3.1 T3 versus 3.1 T4: 62/97 = 63.9%), giving a mean repertoire overlap of 68%. The $\alpha$-diversity of the BKPyV-specific BCR repertoire, representing the total number of BKPyV-specific clonotypes in each patient, was calculated for patients 3.1 and 3.2 using the Chao1 (32) and first-order Jackknife (33) estimators. Similar results were obtained in both patients, with a more precise estimate of 400–500 distinct clonotypes in patient 3.1 (437 ± 25 by Chao1, 449 ± 99 by Jackknife1) compared with 200–700 clonotypes in patient 3.2 (590 ± 99 by Chao1, 304 ± 118 by Jackknife1). Between patients, repertoire overlap was much lower – ~3% (Fig 2H), indicating that each patient's BKPyV-specific BCR repertoire is mostly private, with only a few shared clonotypes between any two individuals. Representative shared clonotypes are shown in Fig 2I. In some cases, distinct light-chain V-genes and/or heavy-chain D-genes were used in the BCRs from different individuals, proving that these cells came from distinct clonal lineages. In other cases, despite identical heavy- and light-chain V(D)J-gene usage, differences in CDR3 length or the number of heavy-chain non-synonymous SHM suggested a distinct clonal history for cells within the same clonotype assigned to different patients. We therefore concluded that these BKPyV-specific BCRs were indeed generated in different individuals, and represented genuine shared or public clonotypes.

Next, we investigated the characteristics of IgM and IgG BKPyV-specific antibodies. As expected, the median number of heavy-chain non-synonymous SHM was lower in B cells expressing IgM or IgD compared with class-switched B cells (Fig 3A). However, there were IgM outliers with very high numbers of SHM, and within

clonotypes that contained both IgG and IgM antibodies, there was no difference in the median number of SHM between IgG and IgM antibodies (Fig 3B). To investigate the evolution of class-switched cells within clonotypes, neighbor-joining trees were constructed for four clonotypes with mixed IgM/IgA/IgG isotypes (Fig S7). Class-switched antibodies did not always cluster together and therefore did not represent a discrete sub-lineage within each clonotype. Rather, it would appear that class switching to IgA or IgG occurs sporadically within memory IgM lineages, as recently described in a longitudinal study of the dynamics of human memory B cells (34).

The relationship between class switching and SHM within clonotypes in BKPyV-specific B cells was analyzed systematically in patients 3.1 and 3.2 by calculating the mean number of heavy-chain SHM and the proportion of class-switched cells for each clonotype (Fig 3C). In both patients, a cluster of expanded IgM clones with low SHM were observed, as well as clonotypes with a progressively higher proportion of class-switched B cells and a progressively higher number of SHM (Fig 3C), culminating in a cluster of clonotypes that were almost exclusively IgG, representing IgG memory B cells (MBG). In addition, both patients also harbored a number of clonotypes with high levels of SHM that were predominantly IgM, presumably IgM memory B cells (MBM). In patient 3.1, the correlation between the number of SHM and the proportion of IgG within clonotypes was highly significant (Fig 3C), whereas in patient 3.2, there was no significant correlation, because of the large number of clonotypes that were predominantly IgM. MBM included the dominant clonotype (number 152) in patient 3.1 (Fig 3C), which had genotype I–specific neutralizing activity when tested on 293TT cells and RS cells (Fig 3F). To confirm that BKPyV replication in KTx recipients induces a serological IgM response, we measured BKPyV-specific IgG and IgM in the three of the patients analyzed by scRNAseq for whom serum was available at the time of KTx, and six additional patients who were seropositive at the time of graft and experienced BKPyV viremia in the first year post-KTx. All nine patients had an increase in BKPyV-specific IgG, and seven of nine also had an increase in BKPyV-specific IgM (Fig S8).

We then compared V-gene usage in BKPyV-specific IgM or IgG antibodies. In both patients 3.1 and 3.2, the IGHV4-39 gene, which showed the greatest enrichment in the overall BKPyV-specific BCR repertoire, was used equally in both IgG and IgM antibodies, whereas several other VH-genes showed preferential use by either IgG or IgM antibodies (Fig 3D and E). In patient 3.1, IGHV1-3, IGHV3-53, IGHV3-66, IGHV4-31, and IGHV4-61 were significantly more frequent in BKPyV-specific IgG compared with IgM, whereas for IGHV1-18, IGHV3-11, IGHV3-48, IGHV3-7, and IGHV5-51, the reverse was true. Although the dataset was smaller, a similar pattern was seen in patient 3.2, with IGHV2-5 and IGHV4-61 more frequently used by BKPyV-specific IgG, whereas IGHV5-51 was more frequent in BKPyV-specific IgM. Significant disparities were also observed in light-chain V-gene usage between BKPyV-specific IgG and IgM (Fig S9). These results indicate that IgG and IgM antibodies specific for BKPyV use overlapping, but not identical, BCR repertoires. To assess the possibility that the presence of non-specific B cells might have skewed the apparent VH-gene usage of BKPyV-specific B cells, we examined the VH-genes present in IgM and IgG cells from the total B cells of patient 3.1 (Fig S10), with specific interest in the two VH-genes that in both patients showed preferential use by BKPyV-specific IgG (IGHV4-61) or IgM (IGHV5-51). In total B cells from patient 3.1, IGHV4-61

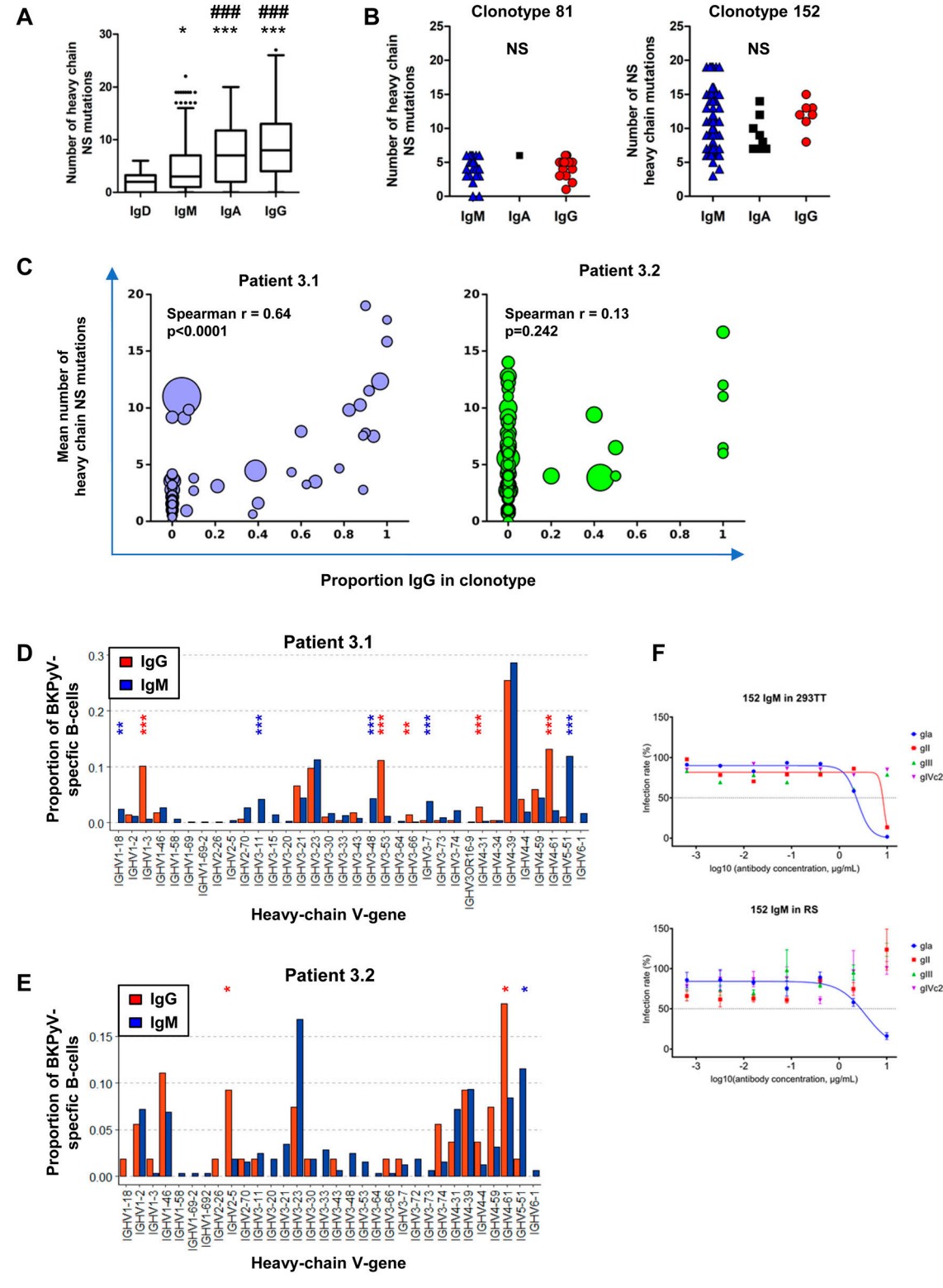

**Figure 3. BKPyV-specific repertoire includes IgM memory B cells with a BCR repertoire distinct from that found in IgG memory B cells.**
**(A)** Number of heavy-chain non-synonymous (NS) mutations in BKPyV-specific B-cell repertoire as a function of antibody isotype. Medians were compared by the Kruskal–Wallis test followed by Dunn's post hoc test. *$P < 0.05$, ***$P < 0.001$ versus IgD; ###$P < 0.001$ versus IgM; IgA versus IgG—not significant. **(B)** Number of heavy-chain NS mutations in IgM, IgA, and IgG antibodies of the same clonotype. Two representative clonotypes are shown. NS—no significant difference in median number of mutations by the Mann–Whitney test (clonotype 81) or the Kruskal–Wallis test (clonotype 152). **(C)** Mean number of heavy-chain NS mutations per clonotype plotted as a function of the proportion of IgG cells within the clonotype. Bubble size is proportional to the number of cells within the clonotype, but the sizes are not normalized

was more frequent in IgM than in IgG ($P < 0.05$, Fisher's exact test), whereas IGHV5-51 was slightly, but not significantly, more frequent in IgG than in IgM (Fig S10). Because these biases are the opposite of what was seen in BKPyV-specific B cells from patient 3.1, contamination with non-specific B cells cannot account for the differences in VH-gene usage observed in BKPyV-specific IgG and IgM isotypes.

## Single-cell RNAseq profiling of BKPyV-specific B cells

Single-cell RNA datasets of TotB and SpecB were integrated with V(D)J datasets to select only B lymphocytes with paired heavy- and light-chain sequences. After filtering to remove low-quality cells, we performed single-cell transcriptome profiling on 5,450 cells, including 3,345 from TotB and 2,105 from SpecB datasets. Using differential expression–based clustering and UMAP visualization, two distinct clusters were observed (Fig 4A). To identify these two clusters, we projected expression levels of *CD27*, a reliable marker of human memory B cells, and IgD isotype, a marker of naive B cells, onto the UMAP plot (Fig 4A). The predominantly IgD⁻CD27⁺ cluster comprised 3,297 memory B cells, whereas the IgD+CD27⁻ cluster comprised 2,153 naive B cells. Furthermore, the naive B-cell cluster showed high expression of *TCL1A* (35), *BACH2* (36, 37, 38), and *CXCR4* (37, 39, 40, 41) (Fig 4B). Concerning the transcriptional signatures of memory B cells, elevated expression of genes associated with the activation of memory B cells and humoral responses was observed, such as the activation marker *CD86* (42, 43), and the *EBI3* (41, 44) gene together with the chemokine receptor *CXCR3* (40, 44) (Fig 4B).

Next, we projected BKPyV-specific IgM and IgG B cells onto UMAP plots (Fig 4C). These cells were mainly found inside the cluster of memory B cells and did not map to distinct sub-clusters within the memory B-cell population. Nevertheless, we tested the hypothesis that there might be specific genes that could distinguish MBM from MBG by analysis of differentially expressed genes using the negative binomial distribution test DESeq2 (45). This showed that the expression profiles of BKPyV-specific MBG and MBM were highly similar, with only 11 genes significantly down-regulated in MBM compared with MBG, and only seven genes, which were significantly up-regulated in MBM (Fig 4D). After *P*-value correction using false discovery rate (46), only *HLA-B* and *HLA-DRB5* genes appeared to be significantly up-regulated in MBG, with adjusted *P*-value < 0.1 (Fig 4D, yellow dots). However, even for these two genes, the log₂ fold changes in gene expression were very low. Furthermore, none of the putatively differentially expressed genes had been identified as markers of specific memory B-cell subsets in previous studies (35, 47, 48). Therefore, we conclude that BKPyV-specific IgM and IgG cells were memory B cells that could not be clearly differentiated by our scRNAseq data.

## Identification of a "41F17-like" family of broadly BKPyV neutralizing antibodies

Broadly neutralizing BKPyV-specific monoclonal antibodies, including the high-affinity clones 41F17 and 27O24, have previously been isolated from the PBMC of a healthy adult donor (26). Because four of the KTx recipients we studied had successfully suppressed BKPyV replication, we investigated whether their BCR repertoire included antibodies with similar broadly neutralizing properties. In order to identify antibodies in our dataset with similar heavy- and light-chain sequences to the 41F17 and 27O24 monoclonals, concatenated CDR3 amino acid sequences were clustered as a phylogenetic tree (Fig 5A).

We were able to identify a cluster of K light-chain IgG antibodies (green clade; Figs 5A and S11) that shared a CDR3 sequence motif with the 41F17 antibody (Fig 5B). When cloned and expressed, antibodies from clonotypes 120, 160, and 198 specifically bound all four genotypes of BKPyV, a property that they shared with the 41F17 antibody, and that was found in only one of nine other antibodies from the dataset that were tested (antibody 206; Fig 6A). Clonotypes 120, 160, and 198 had identical heavy- and light-chain V-gene usage (IGHV4-39/IGKV3-11), which differed from that found in 41F17 (IGHV4-31/IGKV3-11). Interestingly, clonotypes 120, 160, and 198 were characterized by a proportion of >85% IgG within the clonotype, so despite the dominance of IgM in the overall BKPyV-specific repertoire, this cluster of broadly neutralizing antibodies was predominantly IgG.

Binding avidity to VLPs was in the sub-nanomolar range, with antibodies 120 and 160 binding to all four BKPyV genotypes with an avidity similar to that observed for the 41F17 antibody (Fig 5C). In terms of neutralization, like 41F17, antibodies 120, 160, and 198 neutralized all four BKPyV genotypes when assayed in 293TT cells (Fig 5D, upper panels), whereas only antibodies 120, 198, and 41F17 were broadly neutralizing in immortalized renal tubular epithelial cells (Fig 5D, lower panels). The same approach was used to identify antibodies with similar properties to the 27O24 monoclonal; however, although the "nearest-neighbor" IgG antibody in our dataset (clonotype 256) specifically bound to more than one BKPyV genotype, it was not broadly neutralizing (data not shown).

Epitope mapping against a panel of alanine-scanning mutants showed that five of seven genotype I–specific monoclonals bound to the BC-loop of the VP1 protein (Fig 6B). In contrast, binding of antibodies 120, 160, and 198 was not affected by BC-loop mutations, whereas mutation of the E326 residue, which contributes to the 41F17 epitope (26), reduced binding of clonotypes 120 and 160 (Fig 6). To confirm that antibodies 120, 160, and 198 bound to the 41F17 epitope, antibodies were binned by competition ELISA (Fig 7A), which placed antibodies 120, 160, 198, and 206 in the same cluster as 41F17. In contrast, antibodies 256 and 529 clustered with the BC-loop–specific antibody 212. In addition, antibodies 120 and 206 were able to neutralize naturally occurring escape variants (29) (Fig 7B), as well as the genotype I E61K mutant that escapes neutralization by 41F17 (26). However, like 41F17, antibodies 120 and 206 were unable to neutralize genotype IV PSV carrying the E326K mutation. Testing different combinations of monoclonals showed that the combination of antibodies 120 and 304 gave optimal neutralization of BKPyV genotypes and known escape variants (Fig 7C).

---

between patients. Spearman's rank correlation coefficients are shown along with associated *P*-values. **(D, E)** Heavy-chain V-gene usage in SpecB IgG and IgM antibodies in patient 3.1 (D) and patient 3.2 (E). *$P < 0.05$; **$P < 0.01$; ***$P < 0.001$ by Fisher's exact test. Asterisk color denotes the direction of the association: blue—over-represented in IgM antibodies; red—over-represented in IgG antibodies. It should be noted that because of the low number of BKPyV-specific IgG cells in patient 3.2, some differences that appear visually convincing, such as VH3-23, did not reach statistical significance. **(F)** Neutralization of BKPyV PSV by clone 152 expressed as monomeric IgM.

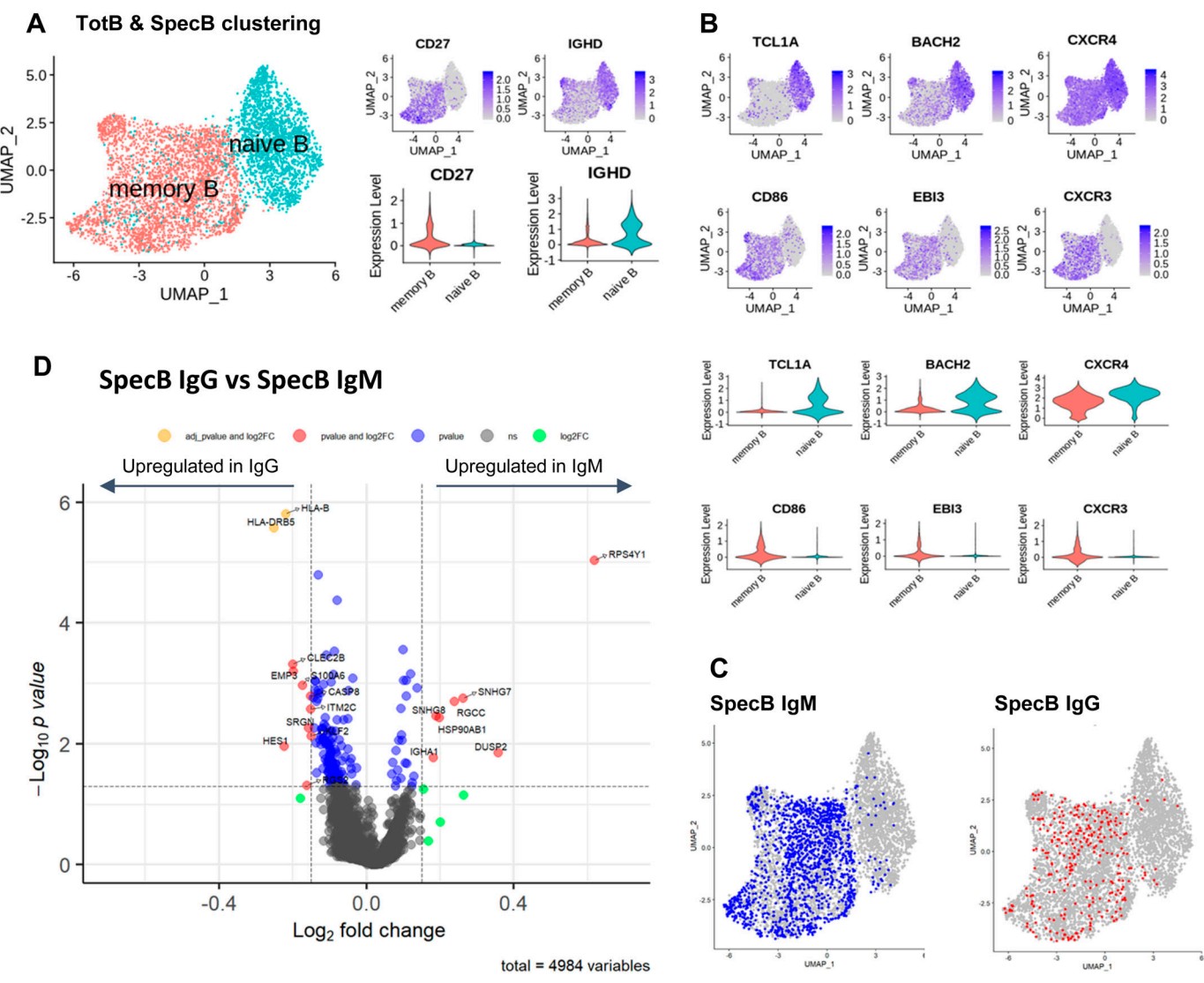

**Figure 4. B-cell clustering and differentially expressed gene analysis of BKPyV-specific B cells.**
**(A)** TotB and SpecB datasets were merged, then clustered based on the differential expression of genes (UMAP). The expression of CD27 and IgD is shown in feature plots (right, upper panels) and in violin plots (right, lower panels). **(B)** Expression levels of specific genes in naive and memory B clusters are displayed in feature plots (six upper panels) and violin plots (six lower panels). **(C)** BKPyV-specific IgM and IgG B cells are projected onto the UMAP plot. **(D)** Volcano plot shows up-regulated genes (right, red dots) and down-regulated genes (left, red and yellow dots) in BKPyV-specific IgM B cells in comparison with BKPyV-specific IgG B cells. Red dots = $P$-value < 0.05. Yellow dots = adjusted $P$-value < 0.1.

## Discussion

The general approach we applied here, sorting antigen-specific B cells, analyzing paired heavy- and light-chain genes by scRNAseq using the 10× Genomics Chromium platform, then producing corresponding antibodies, has previously been used to obtain antibodies against influenza, HIV-1 (49), and SARS-CoV-2 (50, 51). Applying this technique to clinical samples with a reasonable blood draw volume is challenging, because a typical cryopreserved PBMC sample may contain only a few hundred antigen-specific B cells. We therefore used a dual-label hashtagging (52) approach that allowed us to pool 17 different PBMC samples, from which ~$10^4$ BKPyV-specific B cells were sorted, leading to the acquisition of 2,106

paired heavy- and light-chain sequences. This approach makes it possible to obtain the sequences of the totality of antigen-specific B cells, without potential bias of the repertoire after in vitro stimulation, and should in theory allow the production of antibodies from virtually all isolated cells. The main current limitation of our methodology is the cumbersome step of reconstruction of each mAb from the sequences obtained by single-cell RNAseq, which limits the number of antibodies that can be realistically studied. On the contrary, high-throughput B-cell culture systems allow antibody specificity to be determined directly, followed by sequencing to study the repertoire of these antigen-specific B cells (26, 53, 54, 55). However, although efficiency can be high, with up to 50–60% of isolated B cells amplified in vitro, the main challenge is

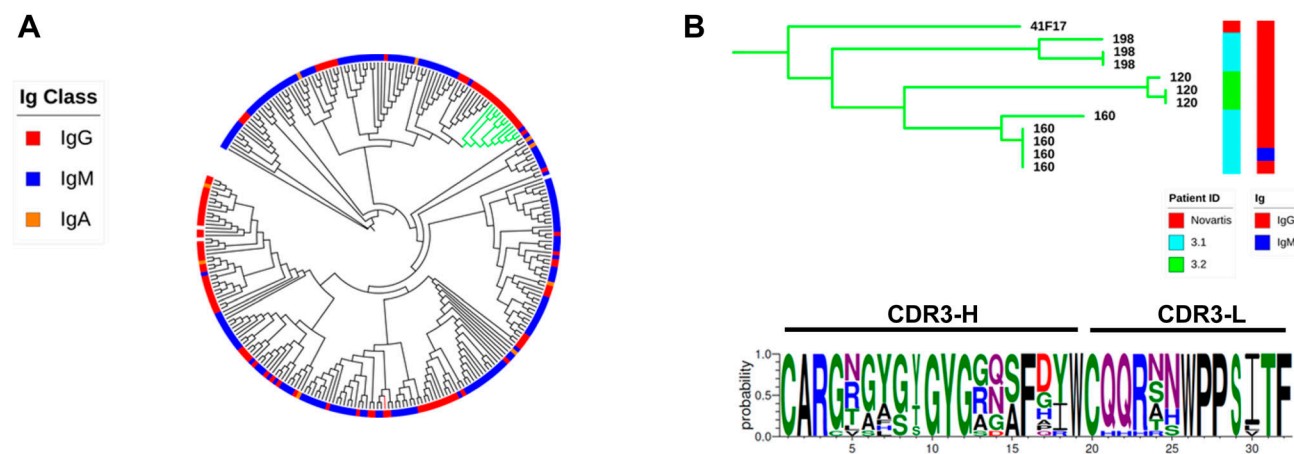

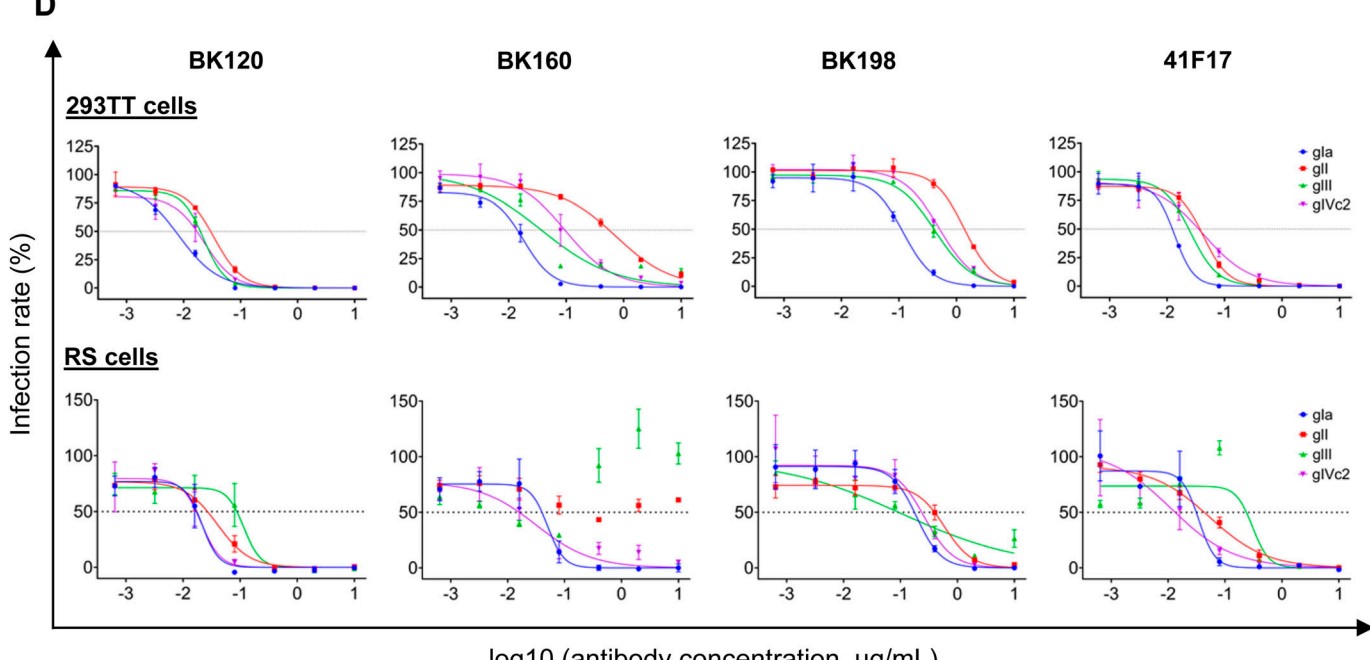

**Figure 5. Clustering of BKPyV-specific IgG and definition of a 41F17-like family of BNAb.**
**(A)** Cladogram of κ light-chain antibodies with at least five somatic hypermutations, clustered on the basis of concatenated CDR3-H and CDR3-L peptide sequences. Green branches show the 41F17-like cluster. The red branch shows the Novartis 27O24 antibody. **(B)** Phylogram of the 41F17-like cluster showing patient ID and antibody isotype, and consensus CDR3-H and CDR3-L sequence characterizing the cluster. **(C)** Binding avidity by SPR and binding EC50 by ELISA for VLPs of different BKPyV genotypes for 41F17-like antibodies. **(D)** Neutralization of BKPyV PSV in 293TT cells and RS cells by 41F17-like antibodies.

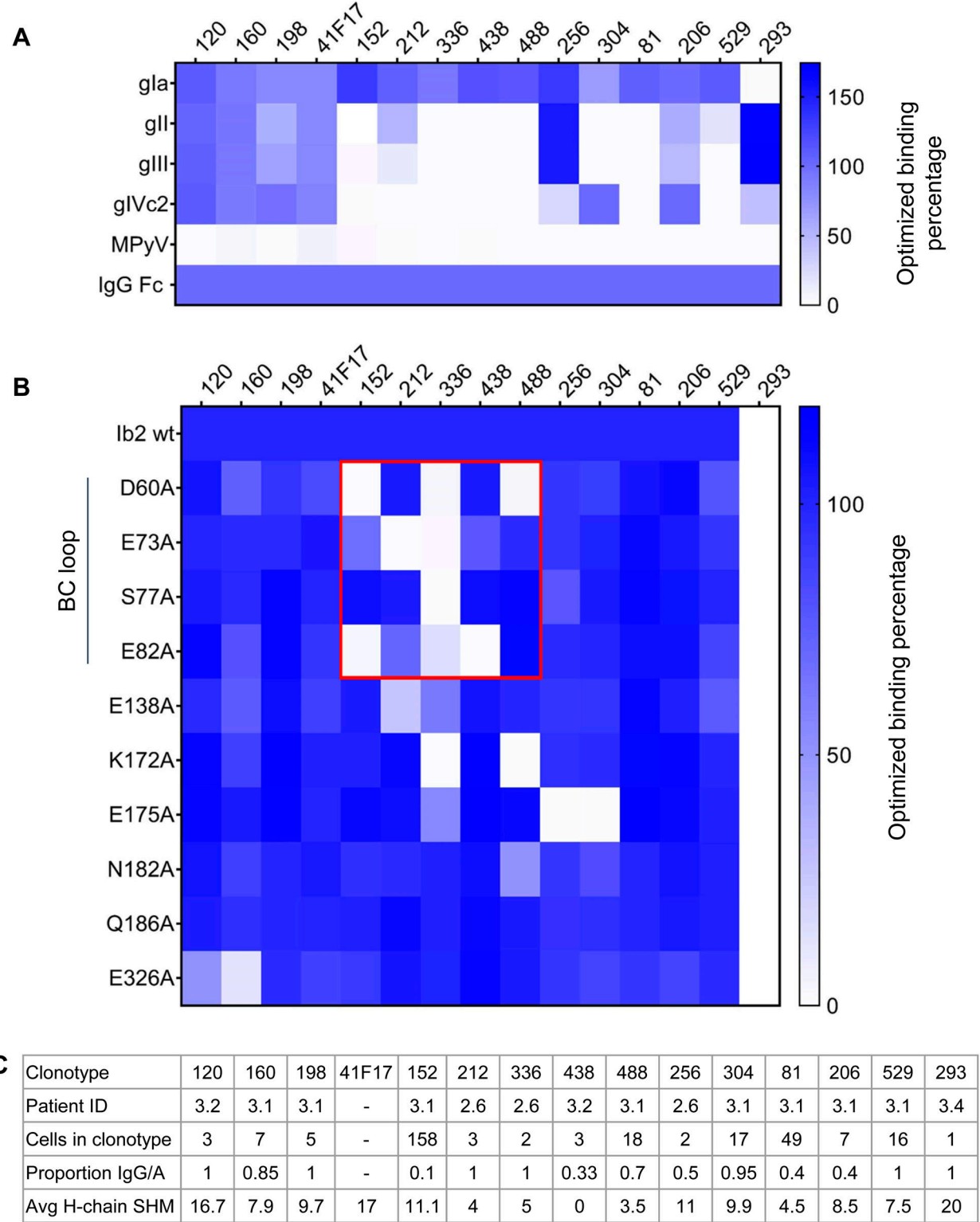

Figure 6.  **Specificity of selected antibodies in the SpecB dataset.**
**(A)** Binding properties of selected antibodies against different BKPyV genotypes and MPyV VLPs. Heatmap indicates ELISA OD450 values normalized to positive control binding to immobilized anti-IgG Fc antibody. **(B)** Binding properties of selected antibodies against genotype Ib2 VLPs carrying alanine-scanning mutations. Antibodies binding to the BC-loop are boxed in red. **(C)** Characteristics of the selected antibody clonotypes.

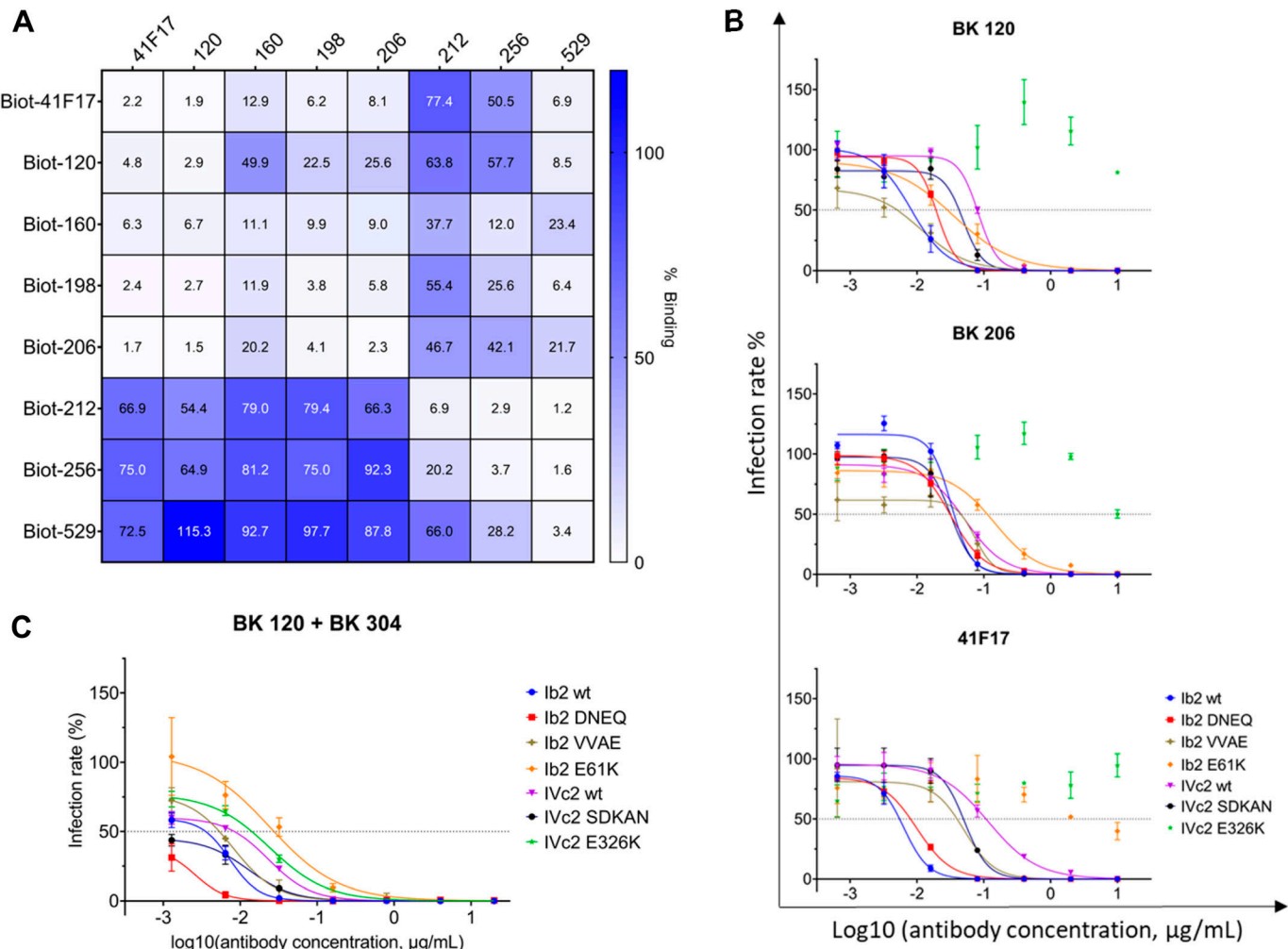

**Figure 7. Epitope binning of selected antibodies, and activity of 41F17-like antibodies against neutralization escape mutants.**
**(A)** Competition ELISA of selected antibodies against BKPyV VLPs. Heatmap indicates the percentage binding of biotinylated antibodies after blocking by the relevant unlabeled antibodies, normalized to the OD450 value observed after binding of the biotinylated antibody in the absence of blocking antibodies. **(B)** Neutralization of VP1 variants by 41F17-like antibodies. Escape mutants observed in patients: DNEQ—K69N and E82Q double mutant; VVAE—L68V, A72V, and E73A triple mutant; and SDKAN—R61S, R69K, E73A, and D77N quadruple mutant. 41F17-resistant variants selected in vitro: E61K; E326K. **(C)** Neutralization of VP1 variants by BK120 and BK304 antibody cocktail.

to process the large number of wells that are seeded. For example, the broadly neutralizing BKV-specific 41F17 antibody was isolated by Lindner et al after screening $1.6 \times 10^5$ clones, which resulted in the identification of ~300 BKPyV-specific antibody sequences (26). In future studies, the optimal strategy may be to combine the advantages of both approaches by sorting antigen-specific B cells, expanding them in short-term culture, then validating antigen specificity in conjunction with high-throughput BCR sequencing using the LIBRA-seq technique (49).

From these sequences, we were able to identify a family of broadly neutralizing BKPyV-specific antibodies, characterized by the use of IGHV4-39 or IGHV4-31 paired with IGKV3-11, with a shared sequence motif in heavy- and light-chain CDR3 regions. These antibodies clustered together with the previously described 41F17 antibody (26) in a competition ELISA, indicating that they bound at, or close to, the 41F17 epitope. Like 41F17, antibodies 120, 160, and 198 were able to neutralize all four BKPyV genotypes, with the highest

neutralizing titer observed for the antibody 120. The three 41F17-like antibodies were identified in BCR sequences from patients 3.1 (160 and 198) and 3.2 (120). Both of these patients had experienced strong BKPyV reactivation post-KTx, followed by long-term control of virus replication subsequent to modulation of immunosuppressive therapy, without the emergence of VP1 neutralization escape mutants. However, because clonotypes 120, 160, and 198 were relatively rare, representing less than 1% of the BCR repertoire in patients 3.1 and 3.2, they could only be detected when several hundred BCR sequences were available. Because far fewer antibody sequences were obtained from patients 3.3 and 3.4, who both experienced persistent high-level BKPyV replication, it was not possible for us to test whether there might be an association between the generation of 41F17-like antibodies and effective control of virus replication in KTx recipients with BKPyV reactivation.

Antibodies 120 and 41F17 retained neutralizing activity against previously described (29) escape mutants in both gI and gIV

backgrounds (Fig 7B), but unlike 41F17, the antibody 120 was resistant to the E61K VP1 mutant. However, both 120 and 41F17 were inactive against PSV with gIV VP1 incorporating a substitution at a different surface-exposed glutamate residue, E326K. Because these PSVs were infectious in both 293TT and RS cells, it is possible that this mutant could emerge in patients, if either 41F17 or 120 were used as monotherapies for the treatment of PyVAN. We therefore screened the different neutralizing antibodies that we had obtained against the E326K mutant to find an optimal combination of two monoclonals that could potentially be combined in an antibody cocktail. The combination of antibodies 120 and 304 gave strong neutralization of all capsid mutants tested, including E326K. In addition, alanine scanning showed that antibody 304 did not bind to the variable BC-loop, but rather to a site that included the VP1 residue 175. This amino acid is not conserved between BKPyV genotypes, showing a D/E/Q polymorphism, and indeed, antibody 304 was not broadly neutralizing (data not shown). However, because antibody 304 was a single clone selected from a group of two closely related clonotypes (304.1 and 304.2) comprising 33 distinct antibodies, it is possible that another clone from this group may show broader neutralization and would therefore be a more effective partner for the antibody 120.

In addition to the identification of broadly neutralizing antibodies, the availability of a large number of sequences in the SpecB dataset made it possible to explore the general properties of the BKPyV-specific BCR repertoire in KTx recipients. Firstly, BKPyV-specific repertoires in all four patients were dominated by lambda light-chain antibodies, with a clear inversion of the kappa/lambda light-chain ratio between TotB and SpecB repertoires. Secondly, although significantly less diverse than the total B-cell repertoires observed in the same patients, BKPyV-specific repertoires were highly varied in all four patients, with many different VH- and VL-genes contributing to the repertoire. Estimates of the overall clonotype diversity obtained in patients 3.1 and 3.2 indicated that an individual's BKPyV-specific BCR repertoire consists of 200–700 distinct clonotypes, with a more precise estimate of 400–500 clonotypes in patient 3.1, no doubt because of the better sampling achieved in this patient. These results imply that accurate measurement of an individual's antibody repertoire against a given antigen requires data from at least two and preferably three independent samples, with BCR sequences obtained for ~500 antigen-specific B cells per sample. The third reproducible feature of the BKPyV-specific BCR repertoire was the high frequency of IgM antibodies, which confirms previous observations in healthy donors (26). Surprisingly, the BCR repertoires mobilized by BKPyV-specific IgM and IgG were not identical, with significant differences in V-gene usage between IgM and IgG antibodies from the same individual (Fig 3D and E).

An important caveat is that our estimates of repertoire diversity rely on the assumption that all the antibody sequences in the SpecB dataset were in fact specific for BKPyV. Unlike in preliminary experiments, where only four of 10 antibodies expressed from sorted B cells were BKPyV-specific (Fig S2), all 10 of the antibodies with no sequence homology to known BKPyV-specific monoclonals that we expressed from the SpecB dataset bound specifically to BKPyV VLP, even when the antibody clonotype was represented by only two or three cells (Fig 6A and C). Furthermore, two different approaches to assess potential contamination of the SpecB dataset

with BCR sequences from non-specific B cells (re-analysis of the sorted cells by flow cytometry, and comparison between the SpecB and TotB repertoires within the same patient) indicated that 90% or more of SpecB sequences were specific. It is likely, however, that a proportion of the singletons (i.e., antibody clonotypes only observed once in the SpecB dataset) are not BKPyV-specific antibodies. This will have a significant impact on estimations of the repertoire size because singletons are crucial for both the Chao1 and Jackknife1 diversity estimators. As the number of non-specific singletons increases, the number of "real" BKPyV-specific singletons decreases, as does the associated repertoire size, so our estimates for the size of the BKPyV-specific repertoire should be considered as an upper limit. In the extreme case, in which all the singleton antibodies are non-specific, the repertoire size in patient 3.1 would be equal to the 192 clonotypes represented by at least two B cells, so we can infer that the lower limit for an individual's BKPyV-specific antibody repertoire is ~200 clonotypes. Our estimate for the size of an individual's BKPyV-specific BCR repertoire (200–700 clonotypes) is therefore robust with respect to the presence of non-specific singleton sequences in the dataset.

Furthermore, our observations concerning the VH-gene usage and the size of the BKPyV-specific antibody repertoire must be interpreted with caution, because they are based on data from only two patients, who shared a rather specific clinical profile: kidney transplant recipients over 60 yr old who were receiving immunosuppressive therapy. This is a major limitation of the present work, which precludes extrapolation of our conclusions to the general population. For example, it is conceivable that the BKPyV-specific antibody repertoire generated in children during primary infection could be very different from what we observed in older KTx recipients. Nevertheless, the high frequency of MBM specific for BKPyV was previously observed in healthy blood donors (26), so this feature of the BKPyV-specific antibody repertoire appears to be a reproducible observation. The specific features of BKPyV-specific MBM therefore warrant further discussion.

Analyzing the proportion of IgG and IgM B cells within clonotypes suggested that BKPyV-specific clonal lineages evolve along two distinct trajectories: firstly, with class switching accompanying SHM, producing clonotypes with progressively greater proportions of IgG and higher levels of SHM, which ultimately lead to clonotypes composed almost exclusively of MBG; and secondly, with SHM accumulating in the absence of class switching, leading to clonotypes that remain predominantly IgM. This may reflect the B-cell activation and differentiation processes occurring in different clonal lineages, with B cells that enter germinal centers giving rise to MBG lineages, whereas those clones that respond to BKPyV capsids in an extrafollicular compartment differentiate into MBM lineages. With this in mind, we sought to test whether BKPyV-specific MBM and MBG displayed differences at the transcriptomic level, but were not able to uncover any convincing differences in gene expression. MBM and MBG did not fall into distinct UMAP clusters, and supervised DGEseq only identified a small number of genes showing quantitatively weak differences in the expression level between MBM and MBG. Our results are therefore consistent with those reported by Sutton et al (47) who found that subsets of circulating B cells defined by scRNAseq did not segregate with antibody isotype.

Why do so many BKPyV-specific B cells differentiate toward MBM? Antibody responses dominated by IgM are characteristic of T-independent antigens (56), and indeed, the humoral response to mouse polyomavirus is largely T-independent (57, 58). Polyomavirus capsids are rigid structures presenting multiple copies of the same epitope in an ordered array, which is typical of TI-2 antigens, and this may favor the generation of a strong MBM response. If this is the case, then BCR repertoires against other non-enveloped icosahedral viruses should also have a strong IgM component, unlike memory B-cell repertoires against the glycoproteins of enveloped viruses, such as HIV, influenza, and SARS-CoV-2, which are predominantly IgG (59, 60, 61).

Looking beyond antiviral immunity, the closest parallel to the results presented here appears to be the humoral response against malarial *Plasmodium* parasites, which includes a substantial MBM component, involving somatically hypermutated IgM in both mice (62) and humans (31, 63). Similar to our results, *P.falciparum*-specific memory IgM contained, on average, fewer SHM than IgG antibodies (31, 63), and in addition, the IgM response appeared to mobilize a distinct BCR repertoire compared with IgG antibodies against the same pathogen (31). It has been suggested that the prevalence of IgM in the *P.falciparum* response may be a consequence of the disruption of germinal center formation during malaria (64), and a similar defect has been observed in KTx recipients, attributed to the inhibitory effects of immunosuppressive therapy on follicular-homing T cells (65). The pre-existing pool of BKV-specific MBM may therefore be preferentially expanded in KTx recipients after BKPyV reactivation, leading to the IgM bias that we observed in our data.

In functional terms, a multimeric IgM specific for the MSP1 antigen, MaliM03, showed much higher avidity for immobilized MSP1 and enhanced parasite inactivation compared with the same antibody expressed as IgG (63), indicating that binding of this IgM to the parasite surface engages multiple paratopes of the IgM, resulting in effective anti-parasite activity. If a similar mechanism pertains to BKPyV neutralization, it would suggest that multimeric BKPyV-specific IgM antibodies may have significant antiviral activity despite the relatively low neutralizing potency that we observed with the clone 152 expressed as monomeric IgG or IgM (Fig 3F). The clone 152 was specific for gI BKPyV, and its binding was sensitive to BC-loop mutations (Fig 6A and B). This region of VP1, which contributes to the sialic acid–binding pocket essential for virus entry, is distributed regularly on the capsid surface in such a way that a pentameric clone 152 IgM could conceivably engage all 10 antibody paratopes (Fig S12). Interestingly, the importance of the geometric distribution of epitopes on the capsid surface for neutralizing potency was recently reported for dengue virus (66).

In contrast, the three 41F17-like clonotypes that we identified were class-switched to IgG, and because the 41F17 epitope is situated at the interface of three VP1 pentamers (26), its geometric distribution on the capsid is not the same as that of the BC-loop. In particular, it would not be possible for a 41F17-like IgM molecule to simultaneously engage all 10 paratopes when binding to BKPyV capsids. Although it is tempting to speculate that epitope specificity may be related to class switching, our data are not consistent with a simple rule such as "BC-loop–specific antibodies remain as IgM," because clonotypes 212, 336, and 488, whose binding was also sensitive to mutations in the BC-loop, were mostly class-switched. Systematic epitope mapping of a larger number of IgM and IgG clonotypes, allied with structural data and bioinformatic modeling, will be required to understand the isotype preference of different antibodies in the BKPyV-specific BCR repertoire.

# Materials and Methods

### Patients and clinical samples

Patients included in this study were transplanted in 2011 and 2012 and had previously been included in a prospective observational study. All patients gave informed consent authorizing the use of archived urine and blood samples for research purposes. Anonymized clinical and biological data for these patients were extracted from the hospital databases. The six patients in this study were diagnosed with BKPyV reactivation based on the detection of viruria >$10^7$ copies/ml correlated with an increase in serum neutralizing titer >1 $\log_{10}$. A total of 17 PBMC samples (two to four per patient, with summary data available in the BioProject https://www.ncbi.nlm.nih.gov/bioproject/PRJNA633999) were collected at different timepoints (1–12 mo) after peak viral load and were cryopreserved in liquid nitrogen.

### Cell culture

HEK293TT cells, purchased from the National Cancer Institute's Developmental Therapeutics Program, were grown in complete DMEM High Glucose (Thermo Fisher Scientific) containing 10% FBS (Dutscher), 100 U/ml penicillin, 100 μg/ml streptomycin (Dutscher), 1× GlutaMAX-I (Thermo Fisher Scientific), and 250 μg/ml hygromycin B (Sigma-Aldrich). RS cells (Evercyte), an immortalized human renal tubule epithelial cell line, were maintained in serum-free medium OptiPRO (Thermo Fisher Scientific) supplemented with 100 U/ml penicillin, 100 μg/ml streptomycin (Thermo Fisher Scientific), and 1× GlutaMAX-I (Thermo Fisher Scientific) in tissue-culture plasticware coated with 50 μg/ml collagen I (Thermo Fisher Scientific). Cells were maintained at 37°C in a humidified 5% $CO_2$ incubator and passaged at confluence by trypsinization for 10 min with 1× TrypLE Express (Thermo Fisher Scientific).

### Plasmids

The BKPyV VP2 and VP3 expression plasmids ph2b (#32109) and ph3b (#32110), and the murine polyomavirus (MPyV) VP1 plasmid (#22519) were purchased from Addgene. VP1 expression plasmids encoding BKPyV genotypes Ia, II, III, and IVc2 were kindly provided by Dr Christopher Buck, National Cancer Institute (NCI). The plasmid pEGFP-N1 (Clontech) was used as the reporter gene. Plasmids containing mutated VP1 sequences have been described previously and were obtained by site-directed mutagenesis using the NEBase changer kit (New England Biolabs) (29).

### Labeled and non-labeled BKPyV VLP production

BKPyV VLPs were prepared following the protocols developed by the Buck laboratory with slight modifications (67). Briefly, 1 × $10^7$

HEK293TT cells were seeded into a 75-cm$^2$ flask in DMEM 10% FBS without antibiotics, then transfected using Lipofectamine 2000 reagent (Invitrogen) according to the manufacturer's instructions. A total of 36 µg VP1 plasmid DNA was mixed with 1.5 ml of Opti-MEM I (Thermo Fisher Scientific). 72 µl of Lipofectamine 2000 was diluted in 1.5 ml of Opti-MEM I and incubated for 5 min at room temperature before mixing with the diluted plasmid DNA. After 20 min at room temperature, 3 ml of DNA–Lipofectamine complexes was added to each flask containing pre-prepared 293TT cells.

Cells were harvested 48 h post-transfection by trypsinization and washed once in PBS and then resuspended in one pellet volume (× µl) of PBS, then mixed with 0.4× µl of 25 U/ml type V neuraminidase (Sigma-Aldrich). After 15 min at 37°C, 0.125× µl of 10% Triton X-100 (Sigma-Aldrich) was added to lyse cells for 15 min at 37°C. The pH of the lysate was adjusted by the addition of 0.075× µl of 1 M ammonium sulfate or sodium bicarbonate; if VLPs were to be fluorescence-labeled before ultracentrifugation, then 1 µl of 250 U/µl Pierce Nuclease (Pierce) was added to degrade free DNA. After 3 h at 37°C, lysates were adjusted to 0.8 M NaCl, incubated on ice for 10 min, and centrifuged at 5,000$g$ for 5 min at 4°C. The supernatant was transferred to a new tube, and the pellet was resuspended in two pellet volumes of PBS/0.8 M NaCl, then centrifuged. The second supernatant was combined with the first, and then, the pooled supernatant was re-clarified by centrifuging. Cleared lysates containing BKPyV genotype I VLPs were labeled with either Alexa Fluor 555 or Alexa Fluor 647, and lysate containing MPyV VLP was labeled with Alexa Fluor 488 using the Alexa Fluor Microscale Protein Labeling kit (Thermo Fisher Scientific).

Labeled or non-labeled lysate was layered onto an OptiPrep 27%/33%/39% gradient (Sigma-Aldrich) prepared in DPBS/0.8 M NaCl, then centrifuged at 175,000$g$ at 4°C overnight in a Sw55TI rotor (Beckman). Tubes were punctured with a 25G syringe needle, and 10 fractions of each gradient were collected into 1.5-ml microcentrifuge tubes. 6.5 µl of each fraction was kept for SDS–PAGE to verify VP1 purity and determine peak fractions for pooling, and then, PBS/5% BSA was added to each fraction to a final concentration of 0.1% BSA as a stabilizing agent. Peak VP1 fractions were pooled, and then, the VP1 concentration of each VLP stock was quantified by migrating 5 µl of the stock on SDS–PAGE, then quantifying the VP1 band by densitometry using a standard curve constructed from a series of fourfold dilutions of BSA starting at 5 µg/well. VLP morphology was confirmed at the electron microscopy facility at the Université François Rabelais, Tours.

### BKPyV pseudovirus (PSV) production

BKPyV PSV particles were prepared following the protocols developed by the Buck laboratory with slight modifications (67). Briefly, cell preparation and transfection were performed similar to BKPyV VLP production. However, instead of transfecting only VP1 plasmid, a total of 36 µg plasmid DNA consisting of 16 µg VP1 plasmid, 4 µg ph2b, 8 µg ph3b, and 8 µg pEGFP-N1 was transfected into 293TT cells.

48h after transfection, producer cells were collected by trypsinization. The pellet was washed once in cold PBS, then resuspended in 800 µl hypotonic lysis buffer containing 25 mM sodium citrate, pH 6.0, 1 mM CaCl$_2$, 1 mM MgCl$_2$, and 5 mM KCl. Cells were subjected to sonication in a Bioruptor Plus device (Diagenode) for 10 min at 4°C with five cycles of 1 min ON/1 min OFF. Type V neuraminidase (Sigma-Aldrich) was added to a final concentration of 1 U/ml and incubated for 30 min at 37°C. 100 µl of 1 M Hepes buffer, pH 7.4 (Thermo Fisher Scientific), was added to neutralize the pH, and then, 1 µl of 250 U/µl Pierce Nuclease (Pierce) was added before incubation for 2 h at 37°C. The lysate was clarified by centrifuging twice at 5,000$g$ for 5 min at 4°C, and PSV was purified in an OptiPrep gradient as described for VLP production. After ultracentrifugation and fraction collection, 8 µl of each fraction was removed for qPCR and the peak fractions were pooled, aliquoted, and stored at –80°C for use in neutralization assays.

For quantification of pEGFP-N1 plasmid, 8 µl of each fraction was mixed with 2 µl of proteinase K buffer containing 100 mM Tris–HCl, pH 7.5 (Thermo Fisher Scientific), 100 mM DTT (Sigma-Aldrich), 25 mM EDTA (Sigma-Aldrich), 1% SDS (Sigma-Aldrich), and 200 µg/ml proteinase K (QIAGEN). This solution was incubated at 50°C for 60 min followed by 95°C for 10 min. Proteinase K extracts were diluted 80-fold in Milli-Q water, and 1 µl was used for qPCR using Applied Biosystems 2× SYBR Green Mix (Applied Biosystems). Primers were CMV-F 5′-CGCAAATGGGCGGTAGGCGTG-3′ and pEGFP-N1-R 5′-GTCCAGCTCGACCAGGATG-3′. Thermal cycling was initiated with a first denaturation step at 95°C for 10 min, followed by 35 cycles of 95°C for 15 s and 55°C for 40 s. Standard curves were constructed using serial dilutions from 10$^7$ to 10$^2$ copies of the pEGFP-N1 plasmid per tube.

### ELISA screening and antibody avidity measurement

Nunc MaxiSorp 96-well plates (Sigma-Aldrich) were coated with 50 ng of BKPyV VLPs in 50 µl PBS at 37°C overnight, then blocked with 5% powdered milk in PBS for 2 h. Supernatants of transfected cells were used at 2 µl per well, and purified antibodies were assayed at fourfold serial dilutions, starting at 10 µg/ml in 50 µl blocking buffer. Antibodies bound to VLPs were detected using goat anti-human IgG horseradish peroxidase–conjugated secondary antibody (Bethyl Laboratories) diluted 1:5,000 in blocking buffer. Washing was performed between each step with PBS/0.05% Tween-20 (Sigma-Aldrich). 50 µl TMB substrate (BD Biosciences) was added, and the reaction was stopped by adding 50 µl of 0.5 M H$_2$SO$_4$. Absorbance was read at 450 nm in a TECAN Spark reader. The effective concentration 50% (EC50) was calculated using GraphPad Prism software.

Competition ELISA experiments were conducted to determine binding epitopes of different generated BKPyV-specific antibodies. Antibodies to test were biotinylated following the manufacturer's instructions (Thermo Fisher Scientific). In brief, 10 ng of BKPyV gI VLPs was coated into ELISA 96-well plates overnight. After washing, the plates were blocked with 5% powdered milk in PBS for 2 h. Non-labeled antibodies were added into wells at 500 ng in 50 µl blocking buffer. After 30-min incubation at 37°C, biotinylated antibodies were added at the EC70 concentration for 1 h. Plates were washed, and specific binding was detected using peroxidase streptavidin (Jackson ImmunoResearch) diluted 1:2,500 in blocking buffer. Absorbance was read at 450 nm. Binding of biotinylated antibodies to BKPyV VLPs in the absence of non-labeled antibodies was used to normalize values.

## Monoclonal antibody binding avidity measurement

The binding strength and kinetics of produced BKPyV-specific antibodies to each BKPyV VLP genotype were assessed using surface plasmon resonance. Briefly, anti-IgG antibody diluted to 20 $\mu$g/ml in sodium acetate buffer, pH 5.0, was immobilized to a CM5 sensor chip (GE Healthcare) until the surface plasmon resonance signal reached 10,000 response units (RUs) using a Biacore T200 (GE Healthcare). BKPyV-specific monoclonal antibodies were then captured at 50 nM concentration. Different BKPyV VLPs including genotypes Ia, II, III, and IVc2 were injected with a series of threefold dilutions from 6 nM to 500 nM. Kinetic parameters and avidity were determined by non-linear regression analysis using Biacore Evaluation software.

## Neutralization assays

293TT and RS cells were seeded at a density of 1 × 10$^4$ cells/well in flat-bottom 96-well plates (BD Falcon), then allowed to attach at 37°C for at least 1 h. BKPyV PSVs were diluted in the corresponding cell culture medium containing antibiotics, 0.1% BSA, and 25 mM Hepes (Thermo Fisher Scientific) to a concentration of 5 x 10$^6$ EGFP-N1 copies/well for RS cells and 2 x 10$^6$ EGFP-N1 copies/well for 293TT cells. Each produced antibody was added into PSV wells to make a series of fivefold antibody dilutions starting at 500 ng/well (i.e., 5 $\mu$g/ml). IgG-PSV mix was incubated at 4°C for 1 h, then added onto plated 293TT or RS cells. Plates were kept in a humidified 5% $CO_2$ incubator at 37°C for 72 h (293TT cells) or 96 h (RS cells). Cells were washed once in PBS/0.5 mM $CaCl_2$ and 0.5 mM $MgCl_2$ before fixing and staining in PBS/0.5 mM $CaCl_2$, 0.5 mM $MgCl_2$, 1% paraformaldehyde (EMS), and 10 $\mu$g/ml Hoechst 33342 (Thermo Fisher Scientific). The number and percentage of GFP$^+$ cells were quantified using a Cellomics ArrayScan VTI HCS Reader (Thermo Fisher Scientific). Neutralization curves were constructed using GraphPad Prism software.

## Cloning, expression, and purification of monoclonal antibodies

Monoclonal antibodies to BKPyV were produced as previously described (68). The selected paired variable heavy (VH)– and variable light (VL)–chain sequences were sent to Eurofins for gene synthesis. VH and VL sequences were cut from the plasmid backbone by restriction enzymes before cloning into expression vectors containing constant regions of heavy chain (C$\gamma$1 of IgG1) and light $\kappa$ chain (C$\kappa$) or light $\lambda$ chain (C$\lambda$). Cloned expression vectors were confirmed by Sanger sequencing, and then, plasmid Max-ipreps were prepared.

Antibodies were first produced on a small scale to check specificity. The day before transfection, 1.5 × 10$^4$ HEK293A cells were seeded into 96-well plates in 200 $\mu$l DMEM supplemented with 1% GlutaMAX and 10% FBS. 125 ng of vH and 125 ng of vL expression vectors were diluted in 25 $\mu$l of 150 mM NaCl, then mixed with 0.5 $\mu$l transfection reagent jetPEI (Polyplus) diluted in 25 $\mu$l of 150 mM NaCl. After 15-min incubation at room temperature, the complex was gently added onto pre-plated 293A cells. 16 h post-transfection, medium was replaced with serum-free medium Pro293a (Lonza) to avoid serum-Igs contamination. Cells were cultured for 5 d at 37°C

in a humidified 5% $CO_2$ incubator, and then, supernatants were harvested and centrifuged at 460$g$ for 5 min to eliminate cells and debris. The presence of antibody in the supernatant was confirmed by ELISA using Affinity-Purified goat anti-human IgG Fc Fragment (BD Biosciences) to coat plates and goat anti-human IgG horseradish peroxidase–conjugated (BD Biosciences) secondary antibody.

Antibodies with confirmed VLP binding were scaled up for production and purification. Briefly, the day before transfection, 6 × 10$^6$ HEK293A cells seeded into 175-cm$^2$ flasks were transfected with 10 $\mu$g of vH and 10 $\mu$g of vL expression vectors following the jetPEI DNA transfection protocol. At 5 d post-transfection, supernatants were harvested and centrifuged at 460$g$ for 5 min to remove cells, then filtered using a 1.22-$\mu$m filter, then a 0.45-$\mu$m filter before purification.

Antibodies were purified using a 1 ml HiTrap rProtein A Fast Flow column (Sigma-Aldrich) on a fast protein liquid chromatography (FPLC) system (Bio-Rad). Firstly, the protein A Sepharose column was equilibrated with 20 mM pH 7.2 phosphate buffer. The filtered supernatant containing antibodies was loaded onto the column, then washed with 20 mM pH 7.2 phosphate buffer, and finally eluted with 0.1 M pH 3 citrate buffer. 500 $\mu$l of each fraction was collected into tubes containing 1 M pH 9 Tris buffer. Optical density was read at 280 nm on a spectrophotometer (Eppendorf) to determine peak fractions to pool. Pooled fractions were dialyzed in a Slide-A-Lyzer cassette (Thermo Fisher Scientific) with a 3.5 K molecular weight cutoff against PBS overnight at 4°C with agitation, then sterilized by filtration at 0.2 $\mu$m. Antibody purity was verified by size-exclusion chromatography using a Superdex 200 Increase 10/300 Gl column (GE Healthcare) following the manufacturer's instructions.

## 10× Chromium single-cell RNA sequencing with immune profiling

### B-cell enrichment from PBMC

B cells were isolated from frozen PBMCs using a human B-cell isolation kit II (Miltenyi Biotec) according to the manufacturer's instructions. Briefly, frozen PBMCs were thawed, counted, and centrifuged, then resuspended in 40 $\mu$l 1× PBS pH 7.2 buffer containing 0.5% BSA and 2 mM EDTA (Fluka), then labeled with a cocktail of biotinylated antibodies targeting non-B cells. Non-B cells were subsequently separated magnetically using the autoMACS Pro Separator with "Deplete" program. Purified untouched B cells were collected and counted before labeling with antibodies and BKPyV VLPs.

### BKPyV-specific B-cell staining and FACS sorting

Enriched B cells from 17 samples were resuspended in 100 $\mu$l of 1× PBS/1% BSA labeling buffer containing the following anti-human antibodies: anti-CD3-BV510 (diluted 1/20) (BD Pharmingen), anti-CD19-BV421 (diluted 1/50) (BD Pharmingen), BKPyV-gI-VLP-Alexa Fluor 555 (1.34 $\mu$g/ml), BKPyV-gI-VLP-Alexa Fluor 647 (0.54 $\mu$g/ml), and MPyV-VLP-Alexa Fluor 488 (1.34 $\mu$g/ml). Patient-specific and timepoint-specific TotalSeq-C oligonucleotide-labeled antibodies (BioLegend) were added into the mix as shown in Fig S4, then incubated at 4°C for 30 min. Cells were washed three times in 13 ml of 1× PBS/1% BSA, and centrifuged at 300$g$ for 5 min at 4°C. After the final wash, cells were resuspended in 1 ml of 1× PBS/0.04% BSA.

10 $\mu$l 7-AAD viability staining solution (BD Pharmingen) was added and incubated for 5 min in the dark. The 17 samples were pooled before sorting through a BD Aria FACS sorter (Becton-Dickinson). 1 × $10^5$ CD19+ B cells were sorted first, followed by BKPyV-specific B cells. The sorted cells were immediately used in the single-cell RNAseq procedure.

### Single-cell 5′ mRNA and VDJ sequencing

CD19+ B cells and BKPyV-specific B cells were loaded onto separate wells of a 10× Chromium A Chip. Subsequent steps were performed using 10× Chromium Single Cell 5′ Library & Gel Bead kit; 10× Chromium Single Cell 5′ Library Construction kit; 10x Chromium Single Cell 5′ Feature Barcode Library kit; 10× Chromium Single Cell V(D)J Enrichment kit, Human B Cell; 10× Chromium Single Cell A Chip kit; and 10× Chromium i7 Multiplex kit according to the manufacturer's protocols. Three libraries were prepared for each cell sample: a V(D)J enriched library, a 5′ gene expression library, and a 5′ cell surface protein library. All libraries were quantified and verified using Quantus (Promega) and Caliper LabChip GX (LifeSciences). Libraries from the same cell sample were pooled at the ratio of 1 V(D)J Enriched: 5 5′ Gene Expression: 1 Cell Surface Protein, and sequenced using an Illumina NextSeq 500 system.

### B-cell single-cell RNAseq data analysis

TotB and SpecB raw sequencing reads in FASTQ files obtained from Illumina sequencing were aligned with the human reference transcriptome (GRCh38) by the 10× Genomics cellranger count pipeline. Output matrices of filtered features of each B population were loaded separately into R version 3.5.1 using the Read10X function. Barcode matrices containing antibody-based hashtag oligonucleotide(HTO) count tables were added into each dataset, followed by a combination of parameters from the TotB and SpecB dataset to create a merged object. $Log_{10}$ HTO counts were graphed to set thresholds for positive and negative labeling by each antibody (Fig S4). SpecB and TotB datasets were analyzed separately, as they were obtained in different sequencing runs, resulting in different HTO count thresholds. In a standard setting, doublets are removed; however, our experiments were designed to obtain double HTO-labeled cells or doublets. Therefore, singlets and cells with wrong HTO combinations were eliminated from the dataset for transcriptome analysis, whereas for BCR repertoire analysis, cells with an unambiguous patient identifier were retained, even if the precise timepoint of the sample could not be determined. Another quality control step was applied to filter out low-quality cells, which had <500 and >3,000 genes detected, and <1,000 and >15,000 RNA counts. Cells with the >10% mitochondrial genes were also excluded. The remaining cells were then joined with the single-cell VDJ data to obtain a merged dataset with productive pairs of heavy- and light-chain sequences in addition to RNAseq read counts. The merged data were used for further gene expression analysis.

Next, immunoglobulin variable genes were deleted from the dataset. To remove batch effects, the combined dataset was processed into the standard workflow of the Seurat Integration and Label Transfer pipeline (69). Raw RNA counts were normalized in a log scale with a factor of 10,000 by default including the percentages of mitochondrial expression and the number of RNA counts as regression factors. Highly variable genes were identified by running FindVariableFeatures functions with the vst selection method, and the results were used as the input for principal component analysis. The RunPCA function returned 30 principal components (PCs) that were used to generate two-dimensional representations via the RunUMAP function. Using the top 15 PCs together with a resolution of 0.1, cells were clustered by computing FindClusters and FindNeighbors function and visualized on a UMAP plot.

### B-cell single-cell VDJ data analysis

Sequencing FASTQ files were submitted to the 10× Genomics cellranger vdj pipeline, the human VDJ reference (GRCh38), for sequence assembly and paired clonotype calling. For further analysis, data were extracted from the all_contig_annotations.json and filtered_contig_annotations.xls output files. Data were rearranged in R and OpenOffice Calc for repertoire analysis using the BRepertoire webtools (70), IMGT-V-Quest (71), and Immunarch (www.immunarch.com). Firstly, clonotypes were identified in BRepertoire based on heavy-chain V- and J-segment usage, then CDR3 homology initially using a Levenshtein distance of 0.20, a slightly more relaxed threshold than the BRepertoire default value of 0.18. Somatic hypermutations (SHM) were identified in heavy-chain V-regions in IMGT-V-Quest, using the cell barcode as the sequence ID, and then, the number of non-synonymous heavy-chain mutations was integrated into the dataframe so that the proportion of IgM clones and mean number of SHM could be analyzed by clonotype using the dplyr package in R. Heavy- and light-chain V-gene usage, repertoire diversity, and repertoire overlap were calculated and visualized using Immunarch. The Chao1 (32) and first-order Jackknife (33) estimators for repertoire diversity were independently calculated using the vegan package in R.

In order to identify antibodies in our dataset with similar heavy- and light-chain sequences to the previously described 41F17 and 27O24 monoclonals (26), antibodies with fewer than five heavy-chain SHM were excluded, and then, the heavy- and light-chain CDR3 amino acid sequences were concatenated and aligned in UGene (72) using a two-step procedure, first with Clustal Ω, then with Muscle to align conserved C-terminal sequences. Aligned sequences were clustered as a phylogenetic tree by neighbor joining. Antibodies with $\kappa$ and $\lambda$ light chains were analyzed separately, and trees were visualized using iTOL 5.7 (https://itol.embl.de/) (73).

### Study approval

Collection of patient samples used in this study was approved by the CHU Nantes local ethics committee on November 8, 2011, and the biobank database was declared to the French Commission Nationale de l'Informatique et des Libertés (n°1600141; CNIL).

## Supplementary Information

## Acknowledgements

This research was funded by grants from the Agence de la Biomedecine (AO Recherche et Greffe 2017), the Fondation Centaure (PAC10, 2017), the Agence Nationale de la Recherche (Project ANR-17-CE17-0003), and the Région Pays de la Loire, project DENISOVIRUS (AO Paris Scientifiques, 2019). Patient recruitment was made possible by funding from the CHU Nantes (AO Interne CHU Nantes 2011). The authors would like to thank Chris Buck for advice in setting up VLP and PSV purification protocols, and Jacques LePendu and Antoine Touzé for discussions at several stages of this work. We are most grateful to the Genomics Core Facility GenoA, a member of Biogenouest and France Genomique, and to the Bioinformatics Core Facility BiRD, a member of Biogenouest and Institut Français de Bioinformatique (IFB) (ANR-11-INBS-0013), for the use of their resources and their technical support. We also acknowledge the IBISA MicroPICell facility (Biogenouest), a member of the national infrastructure France-Bioimaging supported by the French National Research Agency (ANR-10-INBS-04), and Mike Maillasson et, the IMPACT core facility, for SPR data acquisition and analysis.

## Author Contributions

N-K Nguyen: investigation, methodology, and writing—original draft, review, and editing.
M-C Devilder: investigation and methodology.
L Gautreau-Rolland: supervision, investigation, and methodology.
C Fourgeux: investigation.
D Sinha: formal analysis.
J Poschmann: supervision and methodology.
M Hourmant: resources.
C Bressollette-Bodin: conceptualization, resources, and funding acquisition.
X Saulquin: conceptualization, supervision, validation, and writing—review and editing.
D McIlroy: conceptualization, formal analysis, supervision, funding acquisition, investigation, visualization, and writing—original draft, review, and editing.

## Conflict of Interest Statement

The authors declare that they have no conflict of interest.

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
