## [Reviewer comments · Life Science Alliance]

Life Science Alliance

A cluster of broadly neutralizing IgG against BK polyomavirus in a repertoire dominated by IgM

Ngoc-Khanh Nguyen, Marie-Claire DEVILDER, Laetitia Gautreau-Rolland, Cynthia Fourgeux, Debajyoti Sinha, Jeremie Poschmann, Maryvonne Hourmant, Céline Bressollette-Bodin, Xavier saulquin, and Dorian McLroy

DOI: <https://doi.org/10.26508/lsa.202201567>

Corresponding author(s): Dorian McLroy, Nantes Université and Xavier saulquin, CRCNA UMR892

Review Timeline:

Submission Date:	2022-06-20
Editorial Decision:	2022-07-28
Revision Received:	2022-11-10
Editorial Decision:	2022-12-02
Revision Received:	2022-12-18
Editorial Decision:	2022-12-19
Revision Received:	2022-12-30
Accepted:	2023-01-04

Transaction Report:

July 28, 2022

Re: Life Science Alliance manuscript #LSA-2022-01567-T

Dorian McIlroy
UMR1064 INSERM/CRTI
CR2TI, INSERM U1064
NANTES
France

Dear Dr. McIlroy,

Thank you for submitting your manuscript entitled "Overlapping, but distinct, IgM and IgG repertoires in memory B-cells specific for BK polyomavirus" to Life Science Alliance. The manuscript was assessed by expert reviewers, whose comments are appended to this letter. We invite you to submit a revised manuscript addressing the Reviewer comments.

Thank you for this interesting contribution to Life Science Alliance. We are looking forward to receiving your revised manuscript.

Sincerely,

B. MANUSCRIPT ORGANIZATION AND FORMATTING:

Reviewer #1 (Comments to the Authors (Required)):

Summary

Uncontrolled BK polyomavirus (BKPyV) in kidney transplant (KTx) recipients can cause nephropathy (PyVAN) and graft loss or dysfunction. There is no approved antiviral therapy against BKPyV, and ~25% of pts do not clear persistent PyVAN/BKPyV DNAemia with modulation of immunosuppressive therapy. Moreover, persistent PyVAN is associated with increased risk of graft loss. PyVAN is more frequent in KTx with low BKPyV neutralizing antibody titers, but persistent BKPyV can occur despite a robust humoral response - which may be due to the emergence of escape mutations in the viral capsid protein, VP1. The authors hypothesize that differences in BKPyV-humoral responses predispose certain KTx recipients to neutralization escape. Specifically, they hypothesize that KTx recipients with persistent BKPyV DNAemia/PyVAN may have a narrower B cell repertoire or one that may be focused on variable capsid regions rather than conserved epitopes recognized by broadly neutralizing antibodies. Their aim was to characterize features of BKPyV-specific BCR repertoire in KTx recipients who developed a strong humoral response to the virus.

I think this manuscript falls into the scope of Life Science Alliance, as it includes descriptive data of value to the community.

I think the conclusions are overstated in this draft and need to be reworked to better reflect the data.

First, I think the hypothesis is not well supported by the study design, and thus perhaps the introduction could be reworked. The authors are working in the correct patient population (KTx pts with robust anti-BKPyV neutralizing antibody responses); however, they do not have two KTx pt groups- one with persistent BKPyV DNAemia and one without to compare/contrast their B cell repertoires and antibody responses. Second, it would be quite straightforward to compare the breadth of neutralizing antibody responses in sera in two such cohorts, yet they do not even look at the breadth of neutralizing antibody responses in their own pt group. Indeed, 2/ 6 KTx pts in this study had B cells expressing broadly neutralizing antibodies. However, it is not clear if this reflects what is in their sera, particularly as steady state antibody levels derive from long-lived plasma cells in the bone marrow and their monoclonal antibodies derived from memory B cells. Third, they do not return to address their hypotheses in the Discussion. Perhaps because they find B cells that express both broadly neutralizing or non-broadly neutralizing antibodies in the same pts (3.1 and 3.2) and the epitopes of some of the non-broadly neutralizing antibodies are not limited to just the BC loop (e.g., clonotypes 304, 81, 206, and 529).

Major points that need to be addressed

- Cohort size is very small (n = 6), particularly given that many of their conclusions derive from one pt (3.1 or 3.2 depending on the conclusion (i.e., repertoire size vs separate IgM/IgG repertoires, respectively)).
- Discussion - 1st paragraph - "therefore we developed a dual-hashtaging approach" is an overreach and needs to be reworded. Many groups have used cell hashing approaches to combine clinical samples, in fact, this is why the approach was developed (Stoeckius et al. Genome Biology 2018). Cell hashing was not developed by the authors.
- I would remove the Chao1 or Jackknife1 numbers in Fig. 2 and the discussion of these results in the accompanying Results and Discussion sections. Not only are the numbers presented in an unclear way, i.e., chao1 and jackknife1 need to be defined, the table in Fig. 2 needs headers, etc. - these numbers are great odds with the numbers in Fig 2 D-F and are based on very few numbers of clonotypes, therefore the strength of the repertoire size estimation is likely weak. In fact, the authors state in the Discussion that "these results imply that accurate measurement of an individual's antibody repertoire... requires data from.. BCR sequences obtained for ~500 antigen specific B cells per sample" - which was only obtained for pt 3.1 and "Our data are therefore not sufficient to give a valid estimation of this parameter."
- Discussion, page 27 - The authors should delete the statement: "Surprisingly, we were unable to find similar estimates of the size of the human antibody repertoire against any other antigen, and we therefore believe that these are the first measurement of this kind". The authors are definitely not the first group to do this. A quick search easily turned up Chao1 estimates of influenza BCR repertoire diversity in young (n = 10) and elderly vaccinees (n = 10) - Bourcy et al. PNAS 2017. Other groups (e.g., Havenar-Daughton et al Sci Transl Med 2018) have estimated human repertoire size for other antigens (e.g., against HIV vaccine candidates).
- The following statement in the Discussion, page 28, should also be deleted: "however, to the best of our knowledge, the present work is the first time that IgM and IgG repertoires have been compared in antibodies specific for the same antigen". This

is definitely not the first time IgM and IgG repertoires have been compared in antibodies specific for the same antigen (e.g., Durham et al. eLife 2019 PMID 31820734; Kotagiri et al. Cell Reports 2022 PMID 31820734; and many others).

- Page 22, second paragraph: "We then compared V-gene usage in BKPyV-specific MBM and MBG, that is, cells expressing IgM or IgG antibodies with 5 or more NS heavy chain SHM" - Are the authors assuming that B cells must be somatically mutated to recognize, bind or neutralize BKPyV? Do they know that non-somatically mutated (i.e., germline) antibodies cannot recognize, bind or neutralize BKPyV? If they don't know this for certain, this could certainly be biasing the results based on assumptions. There are many viruses where germline antibodies can neutralize - e.g., VSV (Kalinke et al. PNAS 2000 PMID 10963674) and Zika (Magnani et al. PLoS Negl Trop Dis 2017 PMID 10963674) and others. I suggest redoing their analyses to not limit antibodies based on 5 non-synonymous mutations or more. They could also back mutate the monoclonals here to closest germline sequence and test for binding and neutralization.

- Discussion, I find it difficult to conclude that 10 monoclonals expressed from >1675 sequences (<1% of the antibodies sequenced) support "that clonotypes observed at least twice in the SpecB dataset are indeed BKPyV-specific". I think a major limitation of the study is that they did not assess a representative fraction (e.g., 10%) of the antibodies sequenced for specificity by neutralization. It would be acceptable for them to test 10% of the clonotypes but we do not know how many clonotypes there were ultimately (see request for these numbers below).

- Statistics are lacking throughout. For example, Fig 3: can statistical significance be added between groups in A and B? Can correlation be estimated for pt 3.1 and 3.2 in panel C? Can statistical significance be added between IgM vs IgG in panels D and E and in Fig S4?

- Fig 3 panel D and E: There are many V genes where it seems IgM and IgG V gene usages are not wholly different - particularly given the Y-axis scale. For example, in panel D, it seems only ~10% V genes are disparately utilized between IgG and IgM. That is certainly not most. For such a central conclusion from this paper that it ends up in the abstract, and for the concluding Results statement on page 22 ("These results indicate that IgG and IgM antibodies specific for BKPyV use distinct, but somewhat overlapping, BCR repertoires") and page 28 ("Analyzing the proportion of switched and non-switched B-cells within clonotypes showed that BKPyV-specific clonal lineages are either predominantly MBM or MBG, rather than each clonotype comprising (for example) 65% MBM and 35% MBG.") - I would think it should be based on a) more statistical rigor and b) more than 1/6 pts (specifically patient 3.2, panel E). Moreover, it seems the concluding statements on page 22 and page 28 are overstatements and should be re-written to something like, "These results indicate that somatically mutated IgG and IgM antibodies cloned from BKPyV binding B cells frequently utilize the same V genes with differing frequencies."

- The authors use IgD-CD27+ to define memory B cells - what about atypical (IgD-CD27-) memory B cells that are well described in the literature for many pathogens? If I am reading the violin plot correctly, most of the memory B cells are not expressing CD27. Therefore, the authors' analysis should probably be redone to include IgD-CD27- B cells.

- Is the sera of pt 3.1 and 3.2 broadly neutralizing, since they generated broadly neutralizing antibodies? This experiment should be done and results reported if sera are available (from as many time points as available).

- Please state how many clonotypes were identified by sequencing per subject somewhere - ideally in a summary table.

- Discussion, p27 "The third reproducible feature of the BKPyV-specific BCR repertoire was the predominance of IgM antibodies, which confirms previous observations in healthy donors (29)". Where is this shown? Fig 2C shows me that in 3/4 pts there are more class switched SpecB than TotB. This is in contrast to the authors' statement. Predominance means more than. The # of IgM SpecB clonotypes and # IgG SpecB clonotypes per pt should also be clearly shown somewhere - again ideally in a summary table. Is the IgM titer higher than the IgG titer? The titers are also not reported and should be.

- The authors' use of Levenstein distance of 20 for assessing CDRH3 or CDRL3 homology seems quite high - Miho et al. Nat Commun 2019 suggests similar sequences (i.e., clonotypes) have a Levenstein distance of 1-12. The authors' use of 20 could bias the number of identified clonotypes toward the high end - by assuming more antibody sequences are clonotypes than actually are.

Minor points

- Why did the authors not also include a (potentially separate) search for the same V genes in their analysis of antibodies similar to 41F7 and 27O24?

- Page 20, 2nd paragraph "the average repertoire overlap was 68%" I think this is a little misleading and needs to be clarified that this number is the average overlap between any two or three samples for pt 3.1. The average overlap between all three samples is much lower at 36% by my calculations. Similarly - for pt 3.2 - the average overlap is ~32%, which is the number I would report- as it is lower than the stated 51% [calculated from 23/45 clones for T2]. Therefore, it appears that in reality only 1/3 of the clones are shared between any two time points in each patient, which makes much more sense given how little PBMC were collected and how few specific B cells were analyzed.

- In general the measures of diversity and clonal expansion are not well explained in the results, and the authors need to take time to explain, albeit briefly, what each measure (Shannon, D50 index, Gini coefficient) represents and how to interpret the results (e.g., what more or less diversity or more or less clonal expansion looks like for each measure). As written, it seems the Gini coefficient results are giving disparate information to the Shannon and D50 index results ("whereas the unevenness of the repertoire measured by the Gini coefficient (Figure 2F), was significantly higher"), which is not actually the case. Less diversity and more clonal expansion is what is happening and that makes sense if you know what these measures mean.

- Fig S1 the gating is not consistent on BKPyV VLPs - therefore difficult to draw conclusions about frequencies of BKPyV B cells in healthy, seronegative, and seropositive participants (paragraph 2 of Results section).

- Page 10, 1st sentence: "107 to 102 copies" should probably be superscript "7" and "2"

- Page 10, last paragraph, 3rd sentence: what are "BKPyV gla VLPs"? Is "gla" a typo? There are also "BKPyV gl VLPs"

elsewhere. Please define and be consistent.

- Page 18 1st paragraph 1st sentence - I assume this should be "BKPyV *and* murine polyomavirus (MPyV) VLPs..."
- Please define "Frequency" in fig 3 panels D and E and "proportion of cells" in Fig S4 - what's the denominator, respectively? I can assume but it should be defined.
- Fig 3F is in the Figure Legend text but not in the Figure
- Figure 4 has part E which is not included in the figure legend or results.
- Page 23, 1st and 2nd paragraphs: References are provided for CD83, but not TCL1A, BACH2, CXCR4, CD86, EBI3, and CXCR3, CD79B, and PARP1. There is quite a bit of published data on these markers and their expression in naïve and memory B cells and more references should be used to support the authors conclusions.
- Discussion, page 27: "Unlike in preliminary experiments, where only two of seven antibodies expressed from sorted B-cells were BKPyV-specific (Figure 1E)" figure 1E does not show this. The authors should state data not shown or show the data.

Reviewer #2 (Comments to the Authors (Required)):

In this manuscript, the authors describe the sorting and analysis of BK polyomavirus-specific B cells from kidney transplant (KTx) recipients. This study follows-up on a previous work from the group of Traggiai and colleagues performed on healthy individuals in which the BKPyV-specific 41F17 antibody was isolated based on VLPs-based staining of specific B cell and subsequent in vitro B cell cultures (Lindner et al, Immunity 2019). In this study, the authors make similar use of VLPs-based staining, but set-up a different and interesting approach based on hash tagging of enriched B cell from patients to achieve the sorting of total and BKPyV-specific B cells from 6 KTx patients (and 17 total frozen PBMC samples) and paired single-cell RNA and VDJ sequencing using the standard 10X genomics pipeline. This approach allows them to get sufficient cell numbers from two of the patients to perform in depth repertoire analysis and identify the existence of mix unswitched (IgM)/switched (IgG/IgA) clones. Finally, they perform CDR3-based clustering to enrich in silico for putative BCR of interest for the re-expression of broadly neutralizing antibodies and further fully characterize three broadly neutralizing antibodies sharing properties with the previously published 41F17 antibody.

The manuscript provides clearly written introduction and discussion sections. The results section, however, could benefit from some polishing (some erroneous links to key missing figure panels as well as figure panels not necessarily described in the main text, see below) and mostly from being focused on a clearer and better-defined goal for this study. As such, the authors end their introduction on the interesting and highly pertinent hypothesis that characterizing the overall diversity of the repertoire from KTx patients could help identify patients at risk of uncontrolled BKPyV replication. This question, however, doesn't seem to be addressed in the rest of the manuscript. The result section is instead organized in three different sections, one regarding the detailed description of the technological approach used, one regarding the overlapping but distinct IgM/IgG/IgA repertoires and phenotype from two of the patients analyzed (the sole section currently highlighted in the title) and a last and not minor one mostly aimed at identifying broadly neutralizing antibodies in the like of the 41F17 antibody previously described by Traggiai and colleagues, and which mostly ends up selecting IgG clones. No clear link is provided between the last two sections of the work. This overall leaves the reader with the impression of a mostly descriptive and technological rather than a hypothesis-driven work.

Major:

1- One of my major concerns regarding this manuscript lies in the absence of strong validation of the purity of sorted BK-VLP-specific B cells. This point is already discussed by the authors on page 27 of the manuscript. There, the authors mention two sets of experiments (both based on limited sets of re-expressed antibodies) that were performed. One with low purity (2/7 truly BKPyV specific) and listed as being described in Figure 1E but that I could not find there or anywhere else in the manuscript. A similar experiment is described in the result section (page 18) and points to Figure S1 which again seems to be an incorrect link... This should be added in the manuscript. And a second one with better purity (10 out of 10) and listed as being described in Figure 5. Figure 5, however, only show the specificity for 3 antibodies. And it is important to note here that these were not randomly selected antibodies as they were selected from clones clustering with the 41F17 antibodies at the level of the CDR3, a key feature for antigen-specificity as highlighted by the authors themselves.

Achieving above 80% purity isn't necessary for the isolation of antibodies of interest (the goal of the third and last part of this manuscript and indeed the goal of Figure 5). It, however, is essential when one's aims at driving conclusions regarding overall repertoires (second part of this manuscript), as any significant contamination with non-specific (and likely polyclonal) B cells could bias both the estimated diversity as well as the overall IgM/IgG/IgA repartition. It is understood that antibody re-expression is both time-consuming and expensive but other much faster strategies exist such as the in vitro B cell cultures systems set-up and further improved by the groups of Daisuke Kitamura, Garnett Kelsoe or Elisabetta Traggiai.

2- The first part of the results section aims at describing the overall technical strategy chosen by the authors and which they use as one of the selling points for this manuscript. This is indeed an interesting strategy and, although not entirely novel, one that many labs are or will be setting up for the rapid isolation of antibodies of interest against various pathogens. While the authors provide a lot of details, the presentation of such details is sometimes confusing (the overall number of recovered total B cell (3345 listed on Page 22) notably seems quite low with regards to the total number of loaded cells (60.000) and it is unclear if the low numbers of recovered BKPyV-specific B cells from most of the donors (4/6 with less than a hundred) is linked to low numbers of initially sorted cells or to unrelated technical issues). This limits our understanding of the overall efficiency of the

process as regards to other strategies (such as again in vitro B cell culture and/or traditional sanger sequencing). As an example, a simple table recapitulating for each donor the number of total and specific B cell sorted and recovered (with full VDJ) in the scRNA-seq dataset should be included. The authors should also discuss better the benefits and limitations of their approach in regard to previously used approaches.

3- Many figures related to repertoire analysis (including Figure 3C (mix clones) or Figure 4D (differential expression analysis)) oppose the IgM+ cells to pooled IgG+/IgA+ cells. However, the title and abstract only mention IgG and not IgA. The authors should correct this or better explain their rationale.

4- Could the authors explain their rationale behind the cut-off at 5 or more SHM when looking both at the positioning of BKPyV-specific B cells on the UMAP and when performing differential expression analysis between IgM and IgG/IgA BKPyV-specific B cells. It would be interesting to provide the positioning of all BKPyV-specific B cells on the UMAP to get an idea of the frequencies of sorted naive B cells in these cells as well as the possibility that clones of memory B cells with low SHM counts (such as marginal zone B cells) could also be included.

Minor:

1- Figure 2: Label for panel I is currently missing.

2- Figure 3A: Did the author perform any statistical analysis between the groups?

3- Figure 3C: Lineage tree analysis of mixed IgM/IgG/IgA clones could be interesting to understand the evolution within these clonotypes.

4- Figure 4D: CD83 is not visible on this panel although it is one of the main differently expressed gene discussed in the manuscript.

5- Figure 4E: This panel is neither mentioned in the main text nor described in the legend of the figure itself.

6- Figure 6A and C panels do not appear to be described in the main text.

7- Although not available when the manuscript was written, the authors could cite the latest article from the group of Lanzavecchia (<https://doi.org/10.1038/s41590-022-01230-1>) which clearly describe mix IgM/IgG/IgA clone in the human repertoire.

Reviewer #1 (Comments to the Authors (Required)):

First, I think the hypothesis is not well supported by the study design, and thus perhaps the introduction could be reworked. The authors are working in the correct patient population (KTx pts with robust anti-BKPyV neutralizing antibody responses); however, they do not have two KTx pt groups- one with persistent BKPyV DNAemia and one without to compare/contrast their B cell repertoires and antibody responses.

Both Reviewer 1 and Reviewer 2 highlighted this point, and we have therefore revised the Introduction, Discussion, and Abstract, so that the stated aims of the paper are more consistent with the data described.

In particular, we state that the primary aim was to isolate broadly neutralizing antibodies from patients with successful control of BKPyV, and that analysis of the BKPyV-specific BCR repertoire became possible because of the large number of sequences that were obtained. This is a more accurate reflection of the way the project developed over time. In fact, the comparison of BKPyV-specific BCR repertoires between patient groups is the objective of our current work, and in the previous draft, we probably made the mistake of letting our current thinking leak into the Introduction section.

Second, it would be quite straightforward to compare the breadth of neutralizing antibody responses in sera in two such cohorts, yet they do not even look at the breadth of neutralizing antibody responses in their own pt group. Indeed, 2/ 6 KTx pts in this study had B cells expressing broadly neutralizing antibodies. However, it is not clear if this reflects what is in their sera, particularly as steady state antibody levels derive from long-lived plasma cells in the bone marrow and their monoclonal antibodies derived from memory B cells.

Data on neutralizing titres against the four BKPyV genotypes has been added to Figure S3, as detailed below.

Third, they do not return to address their hypotheses in the Discussion.

We hope that reframing the Introduction has resolved this problem.

Major points that need to be addressed

- Cohort size is very small (n = 6), particularly given that many of their conclusions derive from one pt (3.1 or 3.2 depending on the conclusion (i.e., repertoire size vs separate IgM/IgG repertoires, respectively)).

Reviewer 1 is correct that this is the major limitation of the paper, that indeed prevents definitive general conclusions from being drawn. We have highlighted this in the Discussion section.

- Discussion - 1st paragraph - "therefore we developed a dual-hashtaging approach" is an overreach and needs to be re-worded. Many groups have used cell hashing approaches to combine clinical samples, in fact, this is why the approach was developed (Stoeckius et al. Genome Biology 2018). Cell hashing was not developed by the authors.

The sentence has been modified to "therefore we used a dual-hashtaging approach", and the Stoeckius et al. paper has been cited.

For the record, when we designed this scRNAseq experiment in 2019, only 4 TotalSeq-C hashtag antibodies, compatible with BCR sequencing, were commercially available, so in order to multiplex the number of samples we had, we devised a dual-hashtaging approach using combinations of different antibodies to index samples. As far as we are aware, we are still the only group to have done this. That said, the utility of this approach is now rather questionable, since many more TotalSeq-C hashtag antibodies are now available.

- I would remove the Chao1 or Jackknife1 numbers in Fig. 2 and the discussion of these results in the accompanying Results and Discussion sections. Not only are the numbers presented in an unclear way, i.e., chao1 and jackknife1 need to be defined, the table in Fig. 2 needs headers, etc. - these numbers are great odds with the numbers in Fig 2 D-F and are based on very few numbers of clonotypes, therefore the strength of the repertoire size estimation is likely weak. In fact, the authors state in the Discussion that "these results imply that accurate measurement of an individual's antibody repertoire... requires data from.. BCR sequences obtained for ~500 antigen specific B cells per sample" - which was only obtained for pt 3.1 and "Our data are therefore not sufficient to give a valid estimation of this parameter."

Discussion, page 27 - The authors should delete the statement: "Surprisingly, we were unable to find similar estimates of the size of the human antibody repertoire against any other antigen, and we therefore believe that these are the first measurement of this kind". The authors are definitely not the first group to do this. A quick search easily turned up Chao1 estimates of influenza BCR repertoire diversity in young (n = 10) and elderly vaccinees (n = 10) - Bourcy et al. PNAS 2017. Other groups (e.g., Havenar-Daughton et al Sci Transl Med 2018) have estimated human repertoire size for other antigens (e.g., against HIV vaccine candidates).

These two important points are related, and therefore we address them together, starting with the second point. That is, has the size of the human antibody repertoire really been accurately measured for other antigens? We feel that there is currently a real gap in our knowledge concerning this specific question. Broadly speaking, there are two types of paper looking at human BCR repertoires, and the two publications that Reviewer 1 cites are good examples of these two categories.

Bourcy et al. studied the global heavy-chain BCR repertoire in PBMC from young and old individuals before and after influenza vaccination. The Chao1 estimator was used to measure the global BCR repertoire diversity in that study. Specifically, in Figure 2. they show that the global repertoire diversity from a single sample is $\sim 6 \times 10^5$ in young CMV(neg) individuals and $\sim 2 \times 10^5$ in older CMV(neg) individuals.

This is not, however, an accurate estimate of the size of the Influenza-specific antibody repertoire, because the influenza-specific B-cells probably represent only about 1% of the circulating B-cells, and it is impossible to know anything about their repertoire without isolating these rare cells. Well, that's not entirely true, it is possible to identify the major expanded clones after vaccination (or infection), but this restricts the analysis to a small number of immunodominant clones, and therefore represents a very incomplete picture of the overall antigen-specific repertoire. All studies in which antigen-specific B-cells are not specifically identified share this limitation.

The publication from Havenar-Doughton et al. is a good example of the second category. Here, specific B-cells are identified, but the aim of the study was to characterize the properties and frequency of VRC01 class antibodies in the naïve repertoire (answer : 1 in 0.3 million B cells), not to determine the repertoire breadth of the VRC01 class (or other antibodies specific for the HIV gp120 CD4bs epitope) in terms of number of clonotypes, for which no estimate is given in the paper.

In between these two extremes - one type of study looking at global BCR repertoires in PBMC, the other concentrating on a small number of antibody sequences per individual specific for a particular epitope - there is a real need for studies where a sufficient number of antigen-specific B-cells are obtained in order to get a reasonably comprehensive description of the BCR repertoire against that antigen. We sincerely could not find a single paper where this was done, which is why we claimed to be the first group to have generated this type of data.

Now, to return to the first point, Reviewer 1 finds that "these numbers are based on very few numbers of clonotypes, therefore the strength of the repertoire size estimation is likely weak", but the problem with this statement is that the number of clonotypes required to describe the repertoire depends on the size of that repertoire - which is the thing that you don't know in the first place. For example, if the repertoire is really restricted, consisting only of 10 clonotypes, then if you sequence 100 specific B-cells at two different time points, you will only find 10 clonotypes, but this will be a perfect measurement of the size of the repertoire. On the other hand, if the real repertoire is very diverse, say 100 000

clonotypes, and you sequence 1000 specific B-cells at two different time points, you will probably identify something like 2000 distinct clonotypes, but this will still be a very poor estimate of the real repertoire size. So the critical parameter is not the number of clonotypes in the dataset, but rather the number of clonotypes observed relative to the size of the real repertoire.

The problem is that it is not at all obvious how many specific B-cells you need to sequence in order to get a sufficient sampling depth for an accurate measurement to be made, because it depends on the size of the repertoire, which is the unknown quantity that you are trying to measure. Empirically, there are two indications that sampling depth is sufficient: if repeated samples show a high degree of repertoire overlap, and if different approaches to calculate repertoire size give convergent results. Both these criteria were met for patient 3.1, which is why we are convinced that the repertoire size of 400-500 clonotypes for BKPyV-specific antibodies in this patient is accurate. We also believe that our statement, "these results imply that accurate measurement of an individual's antibody repertoire against a given antigen requires data from.. BCR sequences obtained for ~500 antigen specific B cells per sample" is in fact an original result that will help other investigators in this field.

On the other hand, our data do have real weaknesses, because 1/ the proportion of specific antibodies in our dataset was not experimentally validated, and /2 estimates of repertoire size were determined for only two individuals, with a rather specific clinical profile: kidney transplant recipients over 60 years old who were on immunosuppressive therapy. The numbers we obtained cannot therefore be extrapolated to the general population. Both these limitations are discussed in the revised version of the manuscript.

Finally, the sentence quoted by Reviewer 1 "Our data are therefore not sufficient to give a valid estimation of this parameter." referred to the overall repertoire size at the population level, not within an individual.

After this somewhat lengthy discussion of the points raised, what modifications are incorporated into the revised manuscript?

The diversity estimates were removed from Figure 2, as recommended by Reviewer 1, but the Chao1 and Jackknife1 estimates were retained in the text of the Results and Discussion section, with the two major limitations clearly presented. No comment is made on the diversity of the BKPyV-specific repertoire at the population level, and the sentence, "Surprisingly, we were unable to find similar estimates of the size of the human antibody repertoire against any other antigen, and we therefore believe that these are the first measurement of this kind" has been deleted.

- The following statement in the Discussion, page 28, should also be deleted: "however, to the best of our knowledge, the present work is the first time that IgM and IgG repertoires have been compared in antibodies specific for the same antigen". This is definitely not the first time IgM and IgG repertoires have been compared in antibodies specific for the same

antigen (e.g., Durham et al. eLife 2019 PMID 31820734; Kotagiri et al. Cell Reports 2022 PMID 31820734; and many others).

This sentence has been deleted.

- Page 22, second paragraph: "We then compared V-gene usage in BKPyV-specific MBM and MBG, that is, cells expressing IgM or IgG antibodies with 5 or more NS heavy chain SHM" - Are the authors assuming that B cells must be somatically mutated to recognize, bind or neutralize BKPyV? Do they know that non-somatically mutated (i.e., germline) antibodies cannot recognize, bind or neutralize BKPyV? If they don't know this for certain, this could certainly be biasing the results based on assumptions. There are many viruses where germline antibodies can neutralize - e.g., VSV (Kalinke et al. PNAS 2000 PMID 10963674) and Zika (Magnani et al. PLoS Negl Trop Dis 2017 PMID 10963674) and others. I suggest redoing their analyses to not limit antibodies based on 5 non-synonymous mutations or more. They could also back mutate the monoclonals here to closest germline sequence and test for binding and neutralization.

IgM B-cells without SHM may represent new primary responses, in which case, some of these clones may subsequently switch to IgG, while others may remain predominantly IgM. Therefore in order to compare clonal lineages at a similar level of maturation, we restricted the repertoire analysis to IgM or IgG antibodies with 5 or more SHM. This choice was not related to any assumptions concerning binding or neutralizing properties of IgM antibodies.

Nevertheless, as suggested by Reviewer 1, we have repeated the V-gene usage analysis on the IgM and IgG antibody sequences regardless of the extent of SHM, and this has been presented in the revised Figure 3D and 3E. This confirmed the biases in V-gene usage observed in B-cells with 5 or more non-synonymous SHM, and indeed revealed further instances of VH genes that were used almost exclusively by IgM. VH and VL usage was compared between IgG and IgM B-cells with Fisher's exact test, and significant differences are shown in the revised figure.

The DGEseq analysis was also repeated to compare IgG and IgM B-cells regardless of SHM, and these results are presented in the revised Figure 4D. In the revised analysis, only a small number of genes appeared to show differential expression, but this time, with reference to the literature, none of these genes have been reported as showing differential expression between IgM and IgG memory B-cells. We therefore feel that our data should be interpreted with caution, and taking into account that IgM and IgG cells did not form distinct clusters, and that the log₂ fold changes observed were very low (less than 0.4, around 0.2), the most likely conclusion is that BKPyV-specific IgM and IgG cells could not be clearly differentiated by scRNAseq. The text of the Results (p25) and Discussion (p33) sections has therefore been modified, to take this into account.

- Discussion, I find it difficult to conclude that 10 monoclonals expressed from >1675 sequences (<1% of the antibodies sequenced) support "that clonotypes observed at least twice in the SpecB dataset are indeed BKPyV-specific". I think a major limitation of the study is that they did not assess a representative fraction (e.g., 10%) of the antibodies sequenced for specificity by neutralization. It would be acceptable for them to test 10% of the clonotypes but we do not know how many clonotypes there were ultimately (see request for these numbers below).

Reviewer 1 raises a very important point, however, there are two important aspects to our data - generally not available in similar studies - that convinced us of the antigen-specificity of our sequences.

1/ Validation of VLP-binding in sorted B-cells

When individual PBMC samples are sorted with fluorescence-labeled antigens, the number of positive B-cells is so low that the sort gate must be defined before the specific antigen-binding B-cell population is clearly visible in the relevant dot-plots, and this means that it is difficult to completely exclude non-specific B-cells. Furthermore, because the number of sorted cells is so low, there is no way to validate the purity of the sorted cells. Because of this, the proportion of specific B-cells in the sorted population is variable, often considerably less than 100%, and generally not known at the time of cell sorting. This was the case for the preliminary experiments we performed, as shown in the new Figure S2.

However, in our scRNAseq experiment, by pooling a sufficient number of PBMC samples, and enriching B-cells before FACS sorting, we were able to visualize the VLP-binding B-cells clearly, and set the sort gate accurately before starting to sort BKPyV-specific cells. We were also able to re-analyze a small proportion of the sorted cells by flow cytometry, and this data is presented in Figure S5. It shows that ~90% of the SpecB sorted B-cells bound specifically to BKPyV VLPs. Only 76 cells could be analyzed, but this is the equivalent of expressing 76 random antibody sequences from the dataset, and finding that 69 of them bind to BKPyV VLPs.

2/ Comparison with the unsorted BCR repertoire

Since we obtained data on the BCR repertoire in unsorted B-cells, we were able to test directly the possibility that antibody sequences in the SpecB dataset may have been contaminated by sequences from the total B-cell population. In patient 3.1, the TotB dataset contained one dominant clone (ID 4968), represented by 46 cells, and 21 expanded clones, with at least 3 cells in the dataset per clonotype. In the SpecB dataset from the same patient, clone 4968 was not found, and of the 21 expanded clones, only three (clone IDs 152, 81, and 304.1) were represented in the SpecB dataset, where they were among the top 5 expanded clones. The specificity of these three clones was validated experimentally (Figure 6).

The expanded clonotypes in the TotB dataset that were not detected in the SpecB dataset (and therefore non-specific for BKPyV) made up 6.2% of the TotB BCR repertoire (105 cells out of 1691), and this allowed us to estimate the potential contamination of the 1542 SpecB BCR sequences with non-specific sequences. For example, if the SpecB repertoire

contained 10% sequences from total B-cells, then we would expect that 0.6% of the SpecB repertoire, or 10 cells, would carry BCR sequences from the non-specific expanded clonotypes in the TotB repertoire. Since this was significantly different ($p < 0.001$, Fisher's Exact Test) from the observed number (i.e. zero cells), we can therefore conclude that the contamination of the SpecB dataset with non-specific antibody sequences was less than 10%.

The table below shows the calculation for 20%, 10% and 5% contamination:

% Contamination	Number of expected TotB expanded clonotype cells in SpecB dataset	Number of observed TotB expanded clonotype cells in SpecB dataset	p (Fisher exact test, number of expected versus number observed)
20	19	0	<0.001
10	10	0	0.001
5	5	0	0.027

As can be seen from the table, we can have confidence that the contamination of the SpecB dataset with non-specific antibody sequences is less than 10%, and possibly less than 5%, for patient 3.1. As the cells for the other patients were sorted at the same time, this estimate is also valid for the other patients. Indeed, the same analysis carried out on clonotypes found in SpecB and TotB datasets for patient 3.2 gave similar results.

This data has been added as Supplementary Table 3 and described in the Results section (page 21).

- Statistics are lacking throughout. For example, Fig 3: can statistical significance be added between groups in A and B?

Can correlation be estimated for pt 3.1 and 3.2 in panel C?

Can statistical significance be added between IgM vs IgG in panels D and E and in Fig S4?

Statistics have been added to the panels in Fig 3.

- Fig 3 panel D and E: There are many V genes where it seems IgM and IgG V gene usages are not wholly different - particularly given the Y-axis scale. For example, in panel D, it seems only ~10% V genes are disparately utilized between IgG and IgM. That is certainly not most. For such a central conclusion from this paper that it ends up in the abstract, and for the concluding Results statement on page 22 ("These results indicate that IgG and IgM antibodies specific for BKPyV use distinct, but somewhat overlapping, BCR

repertoires") and page 28 ("Analyzing the proportion of switched and non-switched B-cells within clonotypes showed that BKPyV-specific clonal lineages are either predominantly MBM or MBG, rather than each clonotype comprising (for example) 65% MBM and 35% MBG.") - I would think it should be based on a) more statistical rigor and b) more than 1/6 pts (specifically patient 3.2, panel E). Moreover, it seems the concluding statements on page 22 and page 28 are overstatements and should be re-written to something like, "These results indicate that somatically mutated IgG and IgM antibodies cloned from BKPyV binding B cells frequently utilize the same V genes with differing frequencies."

Abstract modified from

"memory IgG and memory IgM B-cells expressed distinct BCR repertoires within the same patient" to

"IgM B-cells in the BKPyV-specific dataset had significant differences in V-gene usage compared to IgG B-cells from the same patient"

Results p22 (revised manuscript p24), revised to "IgG and IgM antibodies specific for BKPyV use overlapping, but not identical BCR repertoires."

Conclusion p28, revised from

"Surprisingly BKPyV-specific IgM used a distinct BCR repertoire compared to that found in class-switched BKPyV-specific antibodies." to,

"Surprisingly, the BCR repertoires mobilized by BKPyV-specific IgM and IgG were not identical, with some significant differences in V-gene usage between IgM and IgG antibodies from the same individual (Figure 3D, 3E)." (p31 revised manuscript)

The statement, "Analyzing the proportion of switched and non-switched B-cells within clonotypes showed that BKPyV-specific clonal lineages are either predominantly MBM or MBG, rather than each clonotype comprising (for example) 65% MBM and 35% MBG." has been replaced with the following text,

"Analyzing the proportion of IgG and IgM B-cells within clonotypes suggested that BKPyV-specific clonal lineages evolve along two distinct trajectories: firstly, with class-switching accompanying SHM, producing clonotypes with progressively greater proportions of IgG and higher levels of SHM, ultimately leading to clonotypes composed almost exclusively of MBG; and secondly, with SHM accumulating in the absence of class-switching, leading to clonotypes that remain predominantly IgM." (p32 revised manuscript)

With respect to the general point that "There are many V genes where it seems IgM and IgG V gene usages are not wholly different.... For example, in panel D, it seems only ~10% V genes are disparately utilized between IgG and IgM. That is certainly not most." Reviewer 1 seems to be suggesting that in order for **any** difference in V-gene usage to be claimed, that **most** V-genes should show disparate usage. This seems to be an unreasonably high bar, since the null hypothesis would be that IgM and IgG against the

same antigen would show exactly the same V-gene usage. We feel that any significant deviation from this is indeed an unexpected finding, and should be highlighted.

Finally, the Reviewer's point that conclusions should be based on, "a) more statistical rigor and b) more than 1/6 pts", is well taken. We have added the statistical significance to Figure 3D and 3E (point a), and while the low number of patients (point b) is the main limitation of the manuscript, it is the case that similar disparities in V-gene usage were seen in both the patients where we had sufficient data. Indeed, the preferential usage of IGHV4-61 by BKPyV-specific IgG, and IGHV5-51 by BKPyV-specific IgM was observed in both patients 3.1 and 3.2.

- The authors use IgD-CD27+ to define memory B cells - what about atypical (IgD-CD27-) memory B cells that are well described in the literature for many pathogens? If I am reading the violin plot correctly, most of the memory B cells are not expressing CD27. Therefore, the authors' analysis should probably be redone to include IgD-CD27- B cells.

The author is correct in understanding that the memory B-cell population contains IgD-CD27- B cells, but the analysis does not need to be redone, because the memory B-cell population is based on the UMAP clusters, not on any one antigen. CD27 expression in naïve and memory populations is shown as confirmation that the memory B-cell population contains cells that express a well characterized memory B-cell marker.

The text in the Results section has been modified to clarify this point,

"To identify these two clusters, we projected expression levels of CD27, a reliable marker of human memory B cells, and IgD isotype, a marker of naive B-cells, onto the UMAP plot (Figure 4A, UMAP). The predominantly IgD-CD27+ cluster comprised 3297 memory B cells, whereas the IgD+CD27- cluster comprised 2153 naive B cells" (p24-25, revised manuscript)

- Is the sera of pt 3.1 and 3.2 broadly neutralizing, since they generated broadly neutralizing antibodies? This experiment should be done and results reported if sera are available (from as many time points as available).

Serum neutralizing titres have been added to Figure S3. Serum was available from M6 and M12 post-KTx for all patients, except for patient 3.2, for whom only M12 serum was available.

- Please state how many clonotypes were identified by sequencing per subject somewhere - ideally in a summary table.

The number of cells and clonotypes in the SpecB and TotB dataset for each patient has been listed in TableS1.

- Discussion, p27 "The third reproducible feature of the BKPyV-specific BCR repertoire was the predominance of IgM antibodies, which confirms previous observations in healthy donors (29)". Where is this shown? Fig 2C shows me that in 3/4 pts there are more class switched SpecB than TotB. This is in contrast to the authors' statement. Predominance

means more than. The # of IgM SpecB clonotypes and # IgG SpecB clonotypes per pt should also be clearly shown somewhere - again ideally in a summary table. Is the IgM titer higher than the IgG titer? The titers are also not reported and should be.

We feel that this is a misunderstanding, probably because Figure 2C was not sufficiently clear. In order to show the data more clearly, this panel has been redrawn to show the proportions of IgD, IgM, IgA and IgG in the total B-cells and BKPyV-specific B-cells of each patient. The accompanying text has been revised to read as follows,

“The proportion of IgD and IgA class-switched antibodies was lower, and the proportion of IgG antibodies was higher in BKPyV-specific B-cells compared to the corresponding total B-cell population. Nevertheless, in all four patients, IgM was the majority isotype in BKPyV-specific as well as total B-cells (Figure 2C)” (p20, revised manuscript)

The number of IgM / IgA / IgG cells is included in Table S1.

Our main objective in performing the ELISAs was to test whether a serum IgM response could be detected in parallel with the expansion of IgM memory B-cells, and we did not determine titres with absolute quantification. We therefore cannot state whether the IgM titre is higher than the IgG titre.

- The authors' use of Levenshtein distance of 20 for assessing CDRH3 or CDRL3 homology seems quite high - Miho et al. Nat Commun 2019 suggests similar sequences (i.e., clonotypes) have a Levenshtein distance of 1-12. The authors' use of 20 could bias the number of identified clonotypes toward the high end - by assuming more antibody sequences are clonotypes than actually are.

The BRepertoire tool that we used (<http://mabra.biomed.kcl.ac.uk/BRepertoire>) suggests a default value of 0.18 for heavy chain sequences, and we therefore used a value close to this recommendation (0.20). It is not clear how these numbers - proposed in the BRepertoire interface, and documentation

(<https://academic.oup.com/nar/article/46/W1/W264/4970507#118388481>) - relate to the “real” Levenshtein distance, which is always an integer. We therefore, perhaps erroneously, assumed that $0.18 = 18$, and that our threshold of 0.20 was equivalent to 20. To clarify this, the Materials and Methods section has been modified to read as follows,

“clonotypes were identified in BRepertoire based on heavy chain V and J segment usage, then CDR3 homology initially using a Levenshtein distance of 0.20, a slightly more relaxed threshold than the BRepertoire default value of 0.18.” (p17, revised manuscript)

On the general point, Reviewer 1 is correct that the number of clonotypes observed depends on the criteria used to define a clonotype, and in clustering by the Levenshtein or Hamming distance, the higher the cutoff, the more sequences are grouped together. However, because SHM automatically increases the sequence diversity within clonotypes, the optimal cutoff will vary depending on the level of SHM in the dataset. Therefore, in exploratory analysis, different values were tested, and results checked by using the L-chain rearrangement as an independent validation of the clonotype designation. With a

handful of exceptions, clonotypes based on the H-chain CDR3 Levenshtein distance of 0.20 shared identical light-chain V-J usage. We noted that these exceptions used closely related light-chain V-genes, and chose to interpret them as falling within the same clonotype.

Also, we checked whether sequences from the TotB data (expected to be non-specific) were clustered with sequences from the SpecB dataset. This led to a higher value of 0.25 being rejected, and the splitting of some clonotypes that were initially identified using a cutoff value of 0.25 (for example, clonotypes 529.1, 529.2, and 529.3). Conversely, if the cutoff was reduced to 0.18, some clonotypes in the SpecB dataset were split, even though they contained BCR that appeared to be clonally related (identical L-chain as well as H-chain V-J usage, and highly similar H- and L-chain CDR3 sequences).

Overall, although our clonotype designation is not perfect, we believe that the errors are marginal, and the potential over-splitting of some clonotypes (for example, clonotypes 529.1, 529.2, and 529.3 all have the same heavy and light-chain rearrangement, and have identical heavy and light-chain CDR3 length, but are split by the Levenshtein threshold of 0.20) is balanced by the grouping of some sequences that are not clonally related (same clonotype based on heavy chain rearrangement and CDR3 similarity, but different light chain rearrangement).

Minor points

- Why did the authors not also include a (potentially separate) search for the same V genes in their analysis of antibodies similar to 41F7 and 27O24?

This was done initially, but there were no hits for 41F17, however, we noticed that there were some clones with almost identical light-chain sequences to 41F17, which also had a similar H-chain CDR3 to 41F17, despite using a different V-gene. This led to the idea that concatenated H- and L- CDR3 sequences might be better.

- Page 20, 2nd paragraph "the average repertoire overlap was 68%" I think this is a little misleading and needs to be clarified that this number is the average overlap between any two or three samples for pt 3.1. The average overlap between all three samples is much lower at 36% by my calculations. Similarly - for pt 3.2 - the average overlap is ~32%, which is the number I would report- as it is lower than the stated 51% [calculated from 23/45 clones for T2]. Therefore, it appears that in reality only 1/3 of the clones are shared between any two time points in each patient, which makes much more sense given how little PBMC were collected and how few specific B cells were analyzed.

The Immunarch suite of tools proposes several measures of repertoire similarity (https://immunarch.com/articles/web_only/v4_overlap.html), of which the simplest is the overlap coefficient -

“overlap coefficient (.method = "overlap") - a normalised measure of overlap similarity. It is defined as the size of the intersection divided by the size of the smaller of the two sets.”

This is the definition we used to calculate the percentage overlap, hence the percentage overlap in patient 3.2 is $23/45 = 51\%$.

The reason why it is not valid to calculate % overlap using the larger sample as the denominator can be seen from the following thought experiment.

Imagine you sequence 10000 antigen-specific B-cells from a single sample, and identify 1000 distinct clonotypes, then from a second independent sample, you only get 100 specific B-cells, with 10 clonotypes, all of which had been identified in the previous sample. In this case, the repertoire overlap is 100%, because all of the clonotypes in the second sample had already been observed in the first sample. The repertoire overlap is not 1% (shared clonotypes divided by the number of clonotypes in the larger sample). Similarly, it is not valid to calculate the mean of these two values, and say that the average overlap is 50.5%.

For patient 3.1, there were three independent samples, so there are three possible two-by-two comparisons, each of which generates a percentage repertoire overlap. The average repertoire overlap for these three samples is the mean of these three percentage values. The numbers are as follows:

3.1 T2 vs 3.1 T3 : $132/177 = 74.6\%$

3.1 T2 vs 3.1 T4 : $64/97 = 66.0\%$

3.1 T3 vs 3.1 T4 : $62/97 = 63.9\%$

The mean of these three values is 68.2%, as stated in the manuscript.

To clarify this point, the text of the Results has been modified to read,

“The BKPyV-specific repertoire overlap found in two samples from patient 3.2, calculated as the number of shared clonotypes divided by the number of clonotypes in the smaller dataset, was 51%, while for the three PBMC samples that provided sufficient data from patient 3.1, three independent two-by-two measures of repertoire overlap were obtained, giving an average repertoire overlap of 68%.” (p21, revised manuscript)

- In general the measures of diversity and clonal expansion are not well explained in the results, and the authors need to take time to explain, albeit briefly, what each measure (Shannon, D50 index, Gini coefficient) represents and how to interpret the results (e.g., what more or less diversity or more or less clonal expansion looks like for each measure). As written, it seems the Gini coefficient results are giving disparate information to the Shannon and D50 index results (“whereas the unevenness of the repertoire measured by the Gini coefficient (Figure 2F), was significantly higher”), which is not actually the case. Less diversity and more clonal expansion is what is happening and that makes sense if you know what these measures mean.

The reviewer’s point is well taken, and the Results section has been modified as follows:

“These characteristics are expected for antigen specific B-cell repertoires: diversity (as measured by the D50 and Shannon indices) is lower, since the total number of clonotypes in the antigen-specific repertoire is lower than in the total B-cell repertoire, while the expansion of dominant clones in an antigen-specific repertoire generates inequality in the repertoire, leading to a higher Gini coefficient.” (p20, revised manuscript)

- Fig S1 the gating is not consistent on BKPyV VLPs - therefore difficult to draw conclusions about frequencies of BKPyV B cells in healthy, seronegative, and seropositive participants (paragraph 2 of Results section).

The figure has been redrawn with identical gates on the different samples.

- Page 10, 1st sentence: "107 to 102 copies" should probably be superscript "7" and "2"

The text has been corrected.

- Page 10, last paragraph, 3rd sentence: what are "BKPyV gla VLPs"? Is "gla" a typo? There are also "BKPyV gl VLPs" elsewhere. Please define and be consistent.

This is not a typo, but refers to the specific BKPyV subtype (Ia) used. The abbreviations "gl" and gIV" have been defined at their first use in the new version of the Introduction, "... neutralization escape mutants in both a genotype I (gl) and genotype IV (gIV) background."

- Page 18 1st paragraph 1st sentence - I assume this should be "BKPyV *and* murine polyomavirus (MPyV) VLPs..."

The reviewer is right, it should have been "BKPyV and murine polyomavirus". The text has been corrected.

- Please define "Frequency" in fig 3 panels D and E and "proportion of cells" in Fig S4 - what's the denominator, respectively? I can assume but it should be defined.

In Figure 3D, 3E, the axis label has been changed to "Proportion of BKPyV-specific B-cells". In FigS4, the following definition has been added to the legend,

““Proportion of cells” = number of cells with given VH gene / total number of cells in the dataset for each patient.”

- Fig 3F is in the Figure Legend text but not in the Figure

- Figure 4 has part E which is not included in the figure legend or results.

This has been corrected; panel 4E has been moved to 3F, where it was supposed to be in the first place.

- Page 23, 1st and 2nd paragraphs: References are provided for CD83, but not TCL1A, BACH2, CXCR4, CD86, EBI3, and CXCR3, CD79B, and PARP1. There is quite a bit of published data on these markers and their expression in naïve and memory B cells and more references should be used to support the authors conclusions.

References to naive and memory B-cell markers have been added (p25, revised manuscript). However, following re-analysis of the data, differential expression of CD79B, CD83 and PARP1 was no longer found between BKPyV-specific IgM and IgG B-cells. Since these genes are no longer discussed, references to them have not been added.

- Discussion, page 27: "Unlike in preliminary experiments, where only two of seven antibodies expressed from sorted B-cells were BKPyV-specific (Figure 1E)" figure 1E does not show this. The authors should state data not shown or show the data.

The figure reference has been corrected to Figure S2, which has been added to the manuscript.

Furthermore, the text has been corrected to read "four of ten antibodies expressed", after the addition of new data on three antibodies from this experiment.

Reviewer #2 (Comments to the Authors (Required)):

General remark:

Both Reviewer 1 and Reviewer 2 highlighted the disconnect between the objectives stated at the end of the Introduction section, and the content of the Results and Discussion. We have therefore revised the Introduction, Discussion, and Abstract, so that the stated aims of the paper are more consistent with the data described.

In particular, we state that the primary aim was to isolate broadly neutralizing antibodies from patients with successful control of BKPyV, and that analysis of the BKPyV-specific BCR repertoire became possible because of the large number of sequences that were obtained. This is a more accurate reflection of the way the project developed over time. In fact, the comparison of BKPyV-specific BCR repertoires between patient groups is the objective of our current work, and in the previous draft, we probably made the mistake of letting our current thinking leak into the Introduction section.

Major:

1- One of my major concerns regarding this manuscript lies in the absence of strong validation of the purity of sorted BK-VLP-specific B cells. This point is already discussed by the authors on page 27 of the manuscript. There, the authors mention two sets of experiments (both based on limited sets of re-expressed antibodies) that were performed. One with low purity (2/7 truly BKPyV specific) and listed as being described in Figure 1E but that I could not find there or anywhere else in the manuscript. A similar experiment is described in the result section (page 18) and points to Figure S1 which again seems to be an incorrect link... This should be added in the manuscript. And a second one with better purity (10 out of 10) and listed as being described in Figure 5. Figure 5, however, only shows the specificity for 3 antibodies. And it is important to note here that these were not randomly selected antibodies as they were selected from clones clustering with the antibodies at the level of the CDR3, a key feature for antigen-specificity as highlighted by the authors themselves.

Achieving above 80% purity isn't necessary for the isolation of antibodies of interest (the goal of the third and last part of this manuscript and indeed the goal of Figure 5). It, however, is essential when one's aims at driving conclusions regarding overall repertoires (second part of this manuscript), as any significant contamination with non-specific (likely polyclonal) B cells could skew IgM/IgG/IgA repartition. It is understood that antibody re-expression is both time-consuming and expensive but other much faster strategies exist such as the in vitro B cell cultures systems set-up and further improved by the groups of Daisuke Kitamura, Garnett Kelsoe or Elisabetta Traggiai.

Reviewer 1 raises a very important point, however, there are two important aspects to our data - generally not available in similar studies - that convinced us of the antigen-specificity of our sequences.

1/ Validation of VLP-binding in sorted B-cells

When individual PBMC samples are sorted with fluorescence-labeled antigens, the number of positive B-cells is so low that the sort gate must be defined before the specific antigen-binding B-cell population is clearly visible in the relevant dot-plots, and this means that it is difficult to completely exclude non-specific B-cells. Furthermore, because the number of sorted cells is so low, there is no way to validate the purity of the sorted cells. Because of this, the proportion of specific B-cells in the sorted population is variable, often considerably less than 100%, and generally not known at the time of cell sorting. This was the case for the preliminary experiments we performed, as shown in the new Figure S2.

However, in our scRNAseq experiment, by pooling a sufficient number of PBMC samples, and enriching B-cells before FACS sorting, we were able to visualize the VLP-binding B-cells clearly, and set the sort gate accurately before starting to sort BkPyV-specific cells. We were also able to re-analyze a small proportion of the sorted cells by flow cytometry, and this data is presented in Figure S5. It shows that ~90% of the SpecB sorted B-cells bound specifically to BkPyV VLPs. Only 76 cells could be analyzed, but this is the equivalent of expressing 76 random antibody sequences from the dataset, and finding that 69 of them bind to BkPyV VLPs.

2/ Comparison with the unsorted BCR repertoire

Since we obtained data on the BCR repertoire in unsorted B-cells, we were able to test directly the possibility that antibody sequences in the SpecB dataset may have been contaminated by sequences from the total B-cell population. In patient 3.1, the TotB dataset contained one dominant clone (ID 4968), represented by 46 cells, and 21 expanded clones, with at least 3 cells in the dataset per clonotype. In the SpecB dataset from the same patient, clone 4968 was not found, and of the 21 expanded clones, only three (clone IDs 152, 81, and 304.1) were represented in the SpecB dataset, where they were among the top 5 expanded clones. The specificity of these three clones was validated experimentally (Figure 6).

The expanded clonotypes in the TotB dataset that were not detected in the SpecB dataset (and therefore non-specific for BkPyV) made up 6.2% of the TotB BCR repertoire (105 cells out of 1691), and this allowed us to estimate the potential contamination of the 1542 SpecB BCR sequences with non-specific sequences. For example, if the SpecB repertoire contained 10% sequences from total B-cells, then we would expect that 0.6% of the SpecB repertoire, or 10 cells, would carry BCR sequences from the non-specific expanded clonotypes in the TotB repertoire. Since this was significantly different ($p < 0.001$, Fisher's Exact Test) from the observed number (i.e. zero cells), we can therefore conclude that the contamination of the SpecB dataset with non-specific antibody sequences was less than 10%.

The table below shows the calculation for 20%, 10% and 5% contamination:

% Contamination	Number of expected TotB expanded clonotype cells in SpecB dataset	Number of observed TotB expanded clonotype cells in SpecB dataset	p (Fisher exact test, number of expected versus number observed)
20	19	0	<0.001
10	10	0	0.001
5	5	0	0.027

As can be seen from the table, we can have confidence that the contamination of the SpecB dataset with non-specific antibody sequences is less than 10%, and possibly less than 5%, for patient 3.1. As the cells for the other patients were sorted at the same time, this estimate is also valid for the other patients. Indeed, the same analysis carried out on clonotypes found in SpecB and TotB datasets for patient 3.2 gave similar results.

This data has been added as Supplementary Table 3 and described in the Results section. (p21, revised manuscript)

2- The first part of the results section aims at describing the overall technical strategy chosen by the authors and which they use as one of the selling points for this manuscript. This is indeed an interesting strategy and, although not entirely novel, one that many labs are or will be setting up for the rapid isolation of antibodies of interest against various pathogens. While the authors provide a lot of details, the presentation of such details is sometimes confusing (the overall number of recovered total B cell (3345 listed on Page 22) notably seems quite low with regards to the total number of loaded cells (60.000) and it is unclear if the low numbers of recovered BKPyV-specific B cells from most of the donors (4/6 with less than a hundred) is linked to low numbers of initially sorted cells or to unrelated technical issues). This limits our understanding of the overall efficiency of the process as regards to other strategies (such as in vitro B cell culture and/or traditional sanger sequencing). As an example, a simple table recapitulating for each donor the number of total and specific B cell sorted and recovered (with full VDJ) in the scRNA-seq dataset should be included.

The authors should also discuss better the benefits and limitations of their approach in regard to previously used approaches.

Summary data has been added in Table S1. It includes the number of PBMC and enriched B-cells recovered from each sample, and a calculation of the percentage yield of both the FACS sort and the 10x Genomics Chromium step. Both these yields were about 20%, which we attribute to the use of frozen cells, with a relatively long procedure after thawing before loading onto the Chromium cassette.

The number of total B-cells for which paired heavy and light chain sequences were obtained was 4591, which was higher than the number of cells which passed the QC threshold for single-cell transcriptome analysis (3345), but this was still a yield of less than 10%. Analysis of hashtag oligonucleotide sequences, and/or the presence of multiple IGH and IGL/IGK sequences associated with the same 10x Genomics barcode showed that the TotB dataset contained many more doublets, and possible triplets, than the SpecB dataset. This led to the exclusion of a large number of cells from the analysis, and therefore lowered the yield. In retrospect, it was an error to overload the Chromium cassette to such an extent.

Strategies for cultivating and differentiating B cells into plasma cells in vitro are one of the methodologies successfully used to obtain Ag-specific antibodies (Lindner et al 2019, Su et al JI 2016 197 10 4163, J Finney, G Kelsoe. JI 2021, Haniuda, Kitamura 2019 Bio protocol). In such strategies, mAb can be obtained from about 50 to 60% of B cells, as about only half of the lymphocytes can be amplified in vitro. Moretheless, variations in yield are observed depending on the isotypes expressed and the naive or memory profile of the isolated cells, as demonstrated in particular by the team of E Traggiai (Lindner et al, 2019). These methods subsequently offer great versatility to assess the antigenic specificity of antibodies, since they are directly produced in the culture supernatant.

Although we cannot determine the yields of mAb generation from sequences obtained in NGS in our study, due to the limited number of antibodies produced, a theoretical efficiency close to 100% should be achievable. The HEK cell will indeed be able to produce most of the Ab naturally expressed by B lymphocytes. This point constitutes one of the major advantages of methodologies based on BCR sequencing at the single cell level since the mAb of all isolated clonotypes, whatever their phenotype, should be produced. The main current limitation is the cumbersome step of reconstruction of each mAb from the sequences obtained by single cell RNA seq. This has limited the number of antibodies generated in our study, limiting the scope of conclusions that can be formally established on the overall quality of the isolated repertoire. This problem could be circumvented by using higher throughput expression and screening methods, such as yeast display or phage display. These elements of comparison between different methods of Ag-specific antibody generation have been added to the beginning of the Discussion. (p28, revised manuscript)

3- Many figures related to repertoire analysis (including Figure 3C (mix clones) or Figure 4D (differential expression analysis)) oppose the IgM+ cells to pooled IgG+/IgA+ cells. However, the title and abstract only mention IgG and not IgA. The authors should correct this or better explain their rationale.

Data has been re-analyzed to compare IgM versus IgG. As shown in the revised Figure 2C, the two main isotypes in the SpecB dataset were IgM and IgG, so this is the relevant comparison to make.

4- Could the authors explain their rationale behind the cut-off at 5 or more SHM when looking both at the positioning of BKPyV-specific B cells on the UMAP and when performing differential expression analysis between IgM and IgG/IgA BKPyV-specific B cells. It would be interesting to provide the positioning of all BKPyV-specific B cells on the UMAP to get an idea of the frequencies of sorted naive B cells in these cells as well as the possibility that clones of memory B cells with low SHM counts (such as marginal zone B cells) could also be included.

IgM B-cells without SHM may represent new primary responses, in which case, some of these clones may subsequently switch to IgG, while others may remain predominantly IgM. Therefore in order to compare clonal lineages at a similar level of maturation, we chose to restrict the repertoire analysis to IgM or IgG antibodies with 5 or more SHM.

Nevertheless, since this point was raised by both Reviewer 1 and Reviewer 2, we have repeated the V-gene usage analysis on the IgM and IgG antibody sequences regardless of the extent of SHM, and this has been presented in the revised Figure 3D and 3E. This confirmed the biases in V-gene usage observed in B-cells with 5 or more non-synonymous SHM, and indeed revealed further instances of VH genes that were used almost exclusively by IgM. VH and VL usage was compared between IgG and IgM B-cells with Fisher's exact test, and significant differences are shown in the revised figure.

The DGEseq analysis was also repeated to compare IgG and IgM B-cells regardless of SHM, and these results are presented in the revised Figure 4D. In the revised analysis, only a small number of genes appeared to show differential expression, but this time, with reference to the literature, none of these genes have been reported as showing differential expression between IgM and IgG memory B-cells. We therefore feel that our data should be interpreted with caution, and taking into account that IgM and IgG cells did not form distinct clusters, and that the log₂ fold changes observed were very low (less than 0.4, around 0.2), the most likely conclusion is that BKPyV-specific IgM and IgG cells could not be clearly differentiated by scRNAseq. The text of the Results (p25) and Discussion (p33) sections has therefore been modified, to take this into account.

Minor:

1- Figure 2: Label for panel I is currently missing.

Has been corrected

2- Figure 3A: Did the author perform any statistical analysis between the groups?

Statistics have been added to Figure 3

3- Figure 3C: Lineage tree analysis of mixed IgM/IgG/IgA clones could be interesting to understand the evolution within these clonotypes.

Neighbour joining trees were constructed for 4 clonotypes with mixed IgM/IgA/IgM isotypes, and are presented in Figure S7. They show that class-switched antibodies do not

cluster together, and therefore do not represent a discrete sub-lineage within the clonotype. Rather, it would appear that class switching to IgA or IgG occurs sporadically within a memory IgM lineage, as described in the paper that Reviewer 2 refers to in point 7. This has been described in the Results section (p22-23, revised manuscript).

4- Figure 4D: CD83 is not visible on this panel although it is one of the main differently expressed gene discussed in the manuscript.

CD83 no longer appears as differentially expressed (see reply to Major point 4)

5- Figure 4E: This panel is neither mentioned in the main text nor described in the legend of the figure itself.

This panel was erroneously referenced in the text as 3F. This has now been corrected, and the Figure Legend modified accordingly.

6- Figure 6A and C panels do not appear to be described in the main text.

Figure 6C

7- Although not available when the manuscript was written, the authors could cite the latest article from the group of Lanzavecchia (<https://doi.org/10.1038/s41590-022-01230-1>) which clearly describe mix IgM/IgG/IgA clone in the human repertoire.

This article has been cited with relation to class switching in the mixed IgM/IgA/IgG clonotypes.

December 2, 2022

Re: Life Science Alliance manuscript #LSA-2022-01567-TR

Dr. Dorian McLroy
Nantes Université
CR2TI, INSERM U1064
CHU Nantes - Hôtel Dieu
NANTES 44000
France

Dear Dr. McLroy,

Thank you for submitting your revised manuscript entitled "A cluster of broadly neutralizing IgG against BK polyomavirus in a repertoire dominated by IgM" to Life Science Alliance. The manuscript has been seen by the original reviewers whose comments are appended below. While the reviewers continue to be overall positive about the work in terms of its suitability for Life Science Alliance, some important issues remain.

Our general policy is that papers are considered through only one revision cycle; however, given that the suggested changes are relatively minor, we are open to one additional short round of revision. Please note that I will expect to make a final decision without additional reviewer input upon re-submission.

Please submit the final revision within one month, along with a letter that includes a point by point response to the remaining reviewer comments.

To upload the revised version of your manuscript, please log in to your account: <https://lsa.msubmit.net/cgi-bin/main.plex>
You will be guided to complete the submission of your revised manuscript and to fill in all necessary information.

B. MANUSCRIPT ORGANIZATION AND FORMATTING:

Sincerely,

Reviewer #1 (Comments to the Authors (Required)):

2nd review thus summary not provided, please see primary review.

Comments:

Overall very responsive to primary review and I think suitable for publication with minor changes highlighted below.

1. Fig 29 discussion (pg 23): I think this discussion needs to be re-worded. I don't think the authors can conclude that Fig S9 supports recall IgM response over what could be a bona fide de novo IgM response, particularly because the IgM response in this manuscript is measured over such long intervals. Recall means a shorter kinetics than during a primary response. Is there data to support that a primary IgM response to BKPyV requires more than 6-12 months to appear?
2. Please label Table S1, S2, and S3
3. Pg 21 line 7 from top: "1542 SpecB BCR Sequences" should this not be "1535" according to Table S1? Also, see 1542 in Table S3. In general, please cross-reference numbers between figures, tables, and text. I think there are several typos - I did not reference all here.
4. Another example: should the numbers in the last paragraph of page 24 match those in Table S1? Also in the 1st paragraph of the Discussion?
5. As class-switched antibodies sometimes cluster together in Fig. S8, I suggest editing text on pg 22 to state "class-switched antibodies did not always cluster together."
6. Pg 23 last paragraph: I think authors omitted calling out IGHV1-18 and IGHV3-66 in the text, which have significance asterisks in Fig. 3D
7. I personally think the two-by-two discussion in the Results Section on pg. 21 could benefit from the addition of the explanation made in the reviewer response, i.e.:

"3.1 T2 vs 3.1 T3 : $132/177 = 74.6\%$; 3.1 T2 vs 3.1 T4 : $64/97 = 66.0\%$; 3.1 T3 vs 3.1 T4 : $62/97 = 63.9\%$ "

Reviewer #2 (Comments to the Authors (Required)):

In this revised manuscript, the authors make substantial changes in the formulation of their working hypothesis and goals. From their initial goal of addressing the repertoire of BKPyV-specific memory B cells (a point that was not strongly supported by the data), they tune down to finding new neutralizing antibodies (a point that is well supported by the latest part of their work). As such the end of the introduction as well as the discussion are clearly improved.

There, however, is still a clear disconnect between carefully worded parts of the introduction and discussion and some of the sentences in the abstract and author summary that should be addressed. It is unclear why the authors would choose to end their abstract by stating that their "results confirm that IgM memory B-cells are a major component of the BKPyV-specific humoral response..." when they do not show any direct activity (neutralizing or other) from memory B cell-derived IgM antibodies in their paper. Similarly, the reference to "distinct repertoires" (point 3, which is not their main goal anymore) and the clone number estimate (point 1, mostly based on one donor) in the author summary should only be kept for the discussion.

Regarding the reasoning of the authors on the specificity of sorted cells in their assays:

1/ Validating the purity of the sort itself, which deals with unstained cells potentially erroneously sorted (new Figure S2), is indeed always an important step. This, however, doesn't tell us much regarding potential non-specific staining (which is likely to vary from one patient to another). The true specificity of sorted memory B cells can only be validated by testing the produced or reexpressed IgM, IgG or IgA by ELISA or other technique...

2/ One of the major hypotheses behind their reasoning that they can rule out potential contaminant by comparing the repertoire of specific B cells and total B cells, is that, if they were to resample twice the total B cell repertoire, they would get an 100% overlap between both sampling. They do not show this and therefore probably overestimate the number of expected TotB clonotypes that they estimate in their table.

The authors should be more careful on their wording regarding these two points in the manuscript (see page 19 or 21 for example)

In the absence of new functional experiments, the authors should limit the overall interpretation that they make of their data to the two points that are reasonably well supported: 1/their characterization of neutralizing anti-BKPyV IgG antibodies (strongly validated) and 2/ the existence of IgM-dominated clones (weaker, but harder to fully explain by simple contamination based on VH usage analysis presented in the manuscript). Regarding the second point, the addition of phylogenetic trees is an interesting supporting point. Two of these trees notably seem to bare the same "numbers" as tested mAbs in figure 3F and 6 (clone 152 and 81). If true (and it seems to be the case based on the discussion on page 34), it would be interesting to show which sequence of the tree was selected for re-expression each time. This would be a strong validation of such IgM-dominated clones.

Minor:

- The author state that their strategy should yield a theoretical efficiency close to 100% as compared to the 50-60% mAbs obtained post culture in systems like the one set-up by the team of E.Traggai. But if you factor in the 20% efficient of cell

encapsulation and sequencing, it seems that culture systems are still more efficient. The authors should reformulate that part of the discussion.

- Figure 6A is still not referred to in the text...

Reviewer #1 (Comments to the Authors (Required)):

2nd review thus summary not provided, please see primary review.

Comments:

Overall very responsive to primary review and I think suitable for publication with minor changes highlighted below.

1. Fig 29 discussion (pg 23): I think this discussion needs to be re-worded. I don't think the authors can conclude that Fig S9 supports recall IgM response over what could be a bona fide de novo IgM response, particularly because the IgM response in this manuscript is measured over such long intervals. Recall means a shorter kinetics than during a primary response. Is there data to support that a primary IgM response to BKPyV requires more than 6-12 months to appear?

Reviewer 1's point is well taken, and the IgM response is probably a mix of reactivation of memory B-cells together with the emergence of new BKPyV-specific clones from the naive pool. The text on p23 has therefore been revised to read,
" To confirm that BKPyV replication in KTx recipients induces a serological IgM response....."
(p23)

2. Please label Table S1, S2, and S3

A header line has been added to the Excel Tables S1, S2, and S3

3. Pg 21 line 7 from top: "1542 SpecB BCR Sequences" should this not be "1535" according to Table S1? Also, see 1542 in Table S3. In general, please cross-reference numbers between figures, tables, and text. I think there are several typos - I did not reference all here.

In Table S1, directly underneath the number 1535 in cell F6, we noted "+7 cells time point undefined", which gives 1542. This is perhaps a rather confusing way to present the data, so to improve clarity, the number of sequences per patient (ex. cell F6) has been corrected to give the total number, including those where it was not possible to assign the time point based on hashtag counts, and the cell directly beneath has been amended to read "Includes 7 cells time point undefined".

4. Another example: should the numbers in the last paragraph of page 24 match those in Table S1? Also in the 1st paragraph of the Discussion?

The numbers cited on p24 are lower because some cells where it was possible to obtain BCR sequences did not pass the filtering criteria for DGEseq analysis. To (hopefully) make this clearer, the sentence has been revised to,

“After filtering to remove low quality cells, we performed single-cell transcriptome profiling on 5450 cells, including 3345 from TotB and 2105 from SpecB datasets.”

The number cited at the beginning of the Discussion was indeed incorrect, and has been corrected to 2106.

5. As class-switched antibodies sometimes cluster together in Fig. S8, I suggest editing text on pg 22 to state "class-switched antibodies did not always cluster together."

Text on p22 has been revised as per the reviewer's suggestion.

6. Pg 23 last paragraph: I think authors omitted calling out IGHV1-18 and IGHV3-66 in the text, which have significance asterisks in Fig. 3D

Reviewer 1 is correct. These V-genes have been added to the text. (p24)

7. I personally think the two-by-two discussion in the Results Section on pg. 21 could benefit from the addition of the explanation made in the reviewer response, i.e.:

"3.1 T2 vs 3.1 T3 : $132/177 = 74.6\%$; 3.1 T2 vs 3.1 T4 : $64/97 = 66.0\%$; 3.1 T3 vs 3.1 T4 : $62/97 = 63.9\%$ "

Text on p21 has been revised as per the reviewer's suggestion.

Reviewer #2 (Comments to the Authors (Required)):

In this revised manuscript, the authors make substantial changes in the formulation of their working hypothesis and goals. From their initial goal of addressing the repertoire of BKPyV-specific memory B cells (a point that was not strongly supported by the data), they tune down to finding new neutralizing antibodies (a point that is well supported by the latest part of their work). As such the end of the introduction as well as the discussion are clearly improved.

There, however, is still a clear disconnect between carefully worded parts of the introduction and discussion and some of the sentences in the abstract and author summary that should be addressed. It is unclear why the authors would choose to end their abstract by stating that their "results confirm that IgM memory B-cells are a major component of the BKPyV-specific humoral response..." when they do not show any direct activity (neutralizing or other) from memory B cell-derived IgM antibodies in their paper. Similarly, the reference to "distinct repertoires" (point 3, which is not their main goal anymore) and the clone number estimate (point 1, mostly based on one donor) in the author summary should only be kept for the discussion.

Two points are made here, which we address in turn.

1/ Last sentence of the abstract.

On reflection, the last sentence of the Abstract was perhaps superfluous. It has been removed in the revised document.

2/ Points highlighted in the Author Summary section.

The Author Summary had indeed remained unchanged from the first version, and we have now updated this section to make it more coherent with the revised manuscript. In particular, the summary of results now reads,

"Here we sequenced antibodies expressed by more than 2000 BK-specific B-cells from kidney transplant recipients, and characterized a family of broadly neutralizing IgG antibodies. We also identified a combination of two monoclonal antibodies that can neutralize previously described BKPyV escape variants. This antibody cocktail may have therapeutic potential for BK polyomavirus associated nephropathy in kidney transplant recipients."

Regarding the reasoning of the authors on the specificity of sorted cells in their assays:

1/ Validating the purity of the sort itself, which deals with unstained cells potentially erroneously sorted (new Figure S2), is indeed always an important step. This, however, doesn't tell us much regarding potential non-specific staining (which is likely to vary from one patient to another). The true specificity of sorted memory B cells can only be validated by testing the produced or reexpressed IgM, IgG or IgA by ELISA or other technique...

As we tried to explain, in this type of experiment it is generally not possible to re-analyze sorted cells, because there are just too few cells per PBMC sample. Therefore, we feel that the statement, “Validating the purity of the sort itself is indeed **always** an important step” (our emphasis) is not an accurate reflection of the state of the art, since it implies that this data is somehow generated as a matter of course, whereas in most cases it is technically impossible to do this. Indeed, sort validation of this kind is not reported in publications where 100-200 B-cells are sorted per sample for precisely this reason. This is why in the general case, when sort purity cannot be verified, it is, as Reviewer 2 rightly states, essential to validate the antigen specificity of the antibodies expressed from sorted cells.

However, our study is unusual (for an antigen-specific BCR/TCR repertoire study) in having sort validation data available, which is why, unlike most other published work in this area, systematic validation of antigen specificity is a little bit less important.

2/ One of the major hypotheses behind their reasoning that they can rule out potential contaminant by comparing the repertoire of specific B cells and total B cells, is that, if they were to resample twice the total B cell repertoire, they would get an 100% overlap between both sampling. They do not show this and therefore probably overestimate the number of expected TotB clonotypes that they estimate in their table.

This is not quite correct. The comparison between the sorted (theoretically BKPyV-specific) and total BCR repertoire assumes that the expanded clones should be present in the same proportion if the total B-cell repertoire were sampled twice, NOT that there would be a 100% overlap. In fact, a 100% overlap is impossible to obtain with a reasonable sampling depth (say a few thousand cells for each sample) because singletons can easily be clones that are present at a vanishingly low frequency in the repertoire, and only show up randomly in any individual sample. That is why our analysis was based only on expanded clonotypes.

Fortunately our data allow us to test this hypothesis (ie. expanded clones should be present in the same proportion in two independent samples of the BCR repertoire) directly, because the expanded clonotypes from patient 3.1 listed in Table S3 are found in PBMC samples from time points T2, T3, and T4 (not possible to test T1, because there are only 36 cells in the data). There are therefore three two-by-two comparisons possible, which give $p=0.198$, $p=0.308$ and $p=1$ by Fisher’s Exact test. The proportion of expanded clones in the total B-cell repertoire data was therefore not significantly different between two independent samples from the same patient, thus validating the hypothesis. The data required for these calculations is provided in Table S2, which includes clonotype ID and patient and timepoint information.

The authors should be more careful on their wording regarding these two points in the manuscript (see page 19 or 21 for example)

The wording on page 19 has been modified as follows,

“ This showed that ~90% of sorted BKPyV-specific B cells stained double positive for AF555 and AF647 labeled BKPyV VLPs, and negative for AF488 labeled MuPyV VLPs.”

Concerning the estimation of the potential contamination by comparison with the total B-cell repertoire, as explained above, we feel that the reviewer's objection is based on a misunderstanding of the hypothesis behind the calculations presented on p21, and that the statistical argument in the current text is valid.

In the absence of new functional experiments, the authors should limit the overall interpretation that they make of their data to the two points that are reasonably well supported: 1/their characterization of neutralizing anti-BKPyV IgG antibodies (strongly validated) and 2/ the existence of IgM-dominated clones (weaker, but harder to fully explain by simple contamination based on VH usage analysis presented in the manuscript). Regarding the second point, the addition of phylogenetic trees is an interesting supporting point. Two of these trees notably seem to bare the same "numbers" as tested mAbs in figure 3F and 6 (clone 152 and 81). If true (and it seems to be the case based on the discussion on page 34), it would be interesting to show which sequence of the tree was selected for re-expression each time. This would be a strong validation of such IgM-dominated clones.

Clonotype numbers are indeed used consistently throughout the text, figures, and supplementary data. Arrows have been added to Figure S8 to show the individual clones that were expressed and tested from clonotypes 81 and 152.

Minor:

- The author state that their strategy should yield a theoretical efficiency close to 100% as compared to the 50-60% mAbs obtained post culture in systems like the one set-up by the team of E.Traggai. But if you factor in the 20% efficient of cell encapsulation and sequencing, it seems that culture systems are still more efficient. The authors should reformulate that part of the discussion.

Reviewer 2 has obliged us to look a little more closely at the papers from the Traggai team, and it appears that the efficiency of B-cell recovery seems to be variable between donors, and can often be much less than the headline-grabbing 50-60% figure. For example, in Lindner et al. Table S2, for total (not antigen-enriched) B-cells, efficiencies were 30-53% in one donor, 15-30% in a second donor, and 8-45% in the third donor, with differences in efficiency between B-cell subsets within the same donor.

However, our intention is not to claim that we have a higher efficiency than screening based on short-term culture, but to contrast the advantages and drawbacks of each approach. Hence the revised text reads as follows, ending with a strategy that could potentially combine the advantages of both approaches.

“On the other hand, high-throughput B cell culture systems allow antibody specificity to be determined directly, followed by sequencing to study the repertoire of these antigen-specific B-cells (60–63). However, although efficiency can be high, with up to 50 to 60% of isolated B cells amplified in vitro, the main challenge is to process the large number of wells that are seeded. For example, the broadly neutralizing BKV-specific 41F17 antibody was isolated by

Lindner et al. after screening 1.6×10^5 clones, which resulted in the identification of approximately 300 BKPyV-specific antibody sequences (60). In future studies, the optimal strategy may be to combine the advantages of both approaches by sorting antigen-specific B-cells, expanding them in short-term culture, then validating antigen-specificity in conjunction with high-throughput BCR sequencing using the LIBRA-seq technique (56).”

- Figure 6A is still not referred to in the text...

The callout to Figure 6A is on p26

“When cloned and expressed, antibodies from clonotypes 120, 160 and 198 specifically bound all four genotypes of BKPyV , a property that they shared with the 41F17 antibody, and which was found in only one of nine other antibodies from the dataset that were tested (antibody 206; Figure 6A)”

This is slightly out of sequence with respect to the text (so should in theory be placed as 5C), but in terms of content, it goes better with the rest of Figure 6, so we would like to request the reviewer’s indulgence on this particular point.

December 19, 2022

RE: Life Science Alliance Manuscript #LSA-2022-01567-TRR

Dr. Dorian McLroy
Nantes Université
CR2TI, INSERM U1064
CHU Nantes - Hôtel Dieu
NANTES 44000
France

Dear Dr. McLroy,

Thank you for submitting your revised manuscript entitled "A cluster of broadly neutralizing IgG against BK polyomavirus in a repertoire dominated by IgM". We would be happy to publish your paper in Life Science Alliance pending final revisions necessary to meet our formatting guidelines.

- please upload both your main and supplementary figures as single files
- please make sure your table files are uploaded as separate editable doc or excel files or make sure they are in the doc file of the manuscript
- please add your supplementary figure legends to the main manuscript text
- please add ORCID ID for secondary corresponding author-you should have received instructions on how to do so
- please consult our manuscript preparation guidelines <https://www.life-science-alliance.org/manuscript-prep> and make sure your manuscript sections are in the correct order
- please use the [10 author names, et al.] format in your references (i.e. limit the author names to the first 10)

A. FINAL FILES:

B. MANUSCRIPT ORGANIZATION AND FORMATTING:

Sincerely,

January 4, 2023

RE: Life Science Alliance Manuscript #LSA-2022-01567-TRRR

Dr. Dorian McLroy
Nantes Université
CR2TI, INSERM U1064
CHU Nantes - Hôtel Dieu
NANTES 44000
France

Dear Dr. McLroy,

Thank you for submitting your Research Article entitled "A cluster of broadly neutralizing IgG against BK polyomavirus in a repertoire dominated by IgM". It is a pleasure to let you know that your manuscript is now accepted for publication in Life Science Alliance. Congratulations on this interesting work.

DISTRIBUTION OF MATERIALS:

Again, congratulations on a very nice paper. I hope you found the review process to be constructive and are pleased with how the manuscript was handled editorially. We look forward to future exciting submissions from your lab.

Sincerely,
